# Epigenetic modifier balances Mapk and Wnt signalling in differentiation of goblet and Paneth cells

Johanna Grinat[1] , Frauke Kosel[1], Neha Goveas[2] , Andrea Kranz[2] , Dimitra Alexopoulou[3] , Klaus Rajewsky[1] , Michael Sigal[4,5], A Francis Stewart[2,6], Julian Heuberger[4,5]

**Differentiation and lineage specification are controlled by co-operation of growth factor signalling. The involvement of epigenetic regulators in lineage specification remains largely elusive. Here, we show that the histone methyltransferase Mll1 prevents intestinal progenitor cells from differentiation, whereas it is also involved in secretory lineage specification of Paneth and goblet cells. Using conditional mutagenesis in mice and intestinal organoids, we demonstrate that loss of Mll1 renders intestinal progenitor cells permissive for Wnt-driven secretory differentiation. However, Mll1-deficient crypt cells fail to segregate Paneth and goblet cell fates. Mll1 deficiency causes Paneth cell-determined crypt progenitors to exhibit goblet cell features by unleashing Mapk signalling, resulting in increased numbers of mixed Paneth/goblet cells. We show that loss of Mll1 abolishes the pro-proliferative effect of Mapk signalling in intestinal progenitor cells and promotes Mapk-induced goblet cell differentiation. Our data uncover Mll1 and its downstream targets Gata4/6 as a regulatory hub of Wnt and Mapk signalling in the control of lineage specification of intestinal secretory Paneth and goblet cells.**

## Introduction

The small intestinal epithelium consists of absorptive enterocytes and secretory cells including Paneth cells, mucus-secreting goblet cells, hormone-producing enteroendocrine cells, and tuft cells, which are organized in crypt-villus structures. Actively proliferating stem cells at the crypt base constantly renew the epithelium (Crosnier et al, 2006; Clevers, 2013). The intestinal stem cells, which are characterized by the expression of the stem cell marker Lgr5 (Barker et al, 2007), give rise to highly proliferative transit-amplifying (TA) progenitor cells, which are located above the stem cell niche in

the crypt. Moving upwards towards the crypt-villus junction, the TA cells differentiate into absorptive enterocytes or secretory cells. The differentiating cells migrate into the villi, except for the Paneth cells which remain at the bottom of the crypts, interspersed between the Lgr5+ stem cells, secrete antimicrobial peptides such as defensins and lysozyme, and provide Wnt ligands and other growth factors for stem cell maintenance (Sato et al, 2011).

In recent years, research generated insights into a network of transcription factors and signalling pathways that guide cell type specification in the TA zone (Beumer & Clevers, 2021). Active Wnt signalling is critical for intestinal homeostasis, stem cell maintenance, and the formation of secretory progenitor cells (Pinto et al, 2003; Fevr et al, 2007). The transcription factor Math1 (Atoh1) is essential for specification towards the secretory lineage (Yang et al, 2001). Gfi1, a transcriptional repressor of enteroendocrine specification, acts downstream of Math1 to select Paneth/goblet cell fates versus the enteroendocrine fate (Shroyer et al, 2005). The ETS transcription factor Spdef further directs secretory maturation in Paneth and goblet cells and deletion of *Spdef* in the mouse intestine leads to an accumulation of immature secretory progenitor cells (Gregorieff et al, 2009; Noah, 2010). Persisting high Wnt activity in secretory progenitor cells promotes the differentiation of Paneth cells (van Es et al, 2005). Paneth cells are absent in mice deficient for Tcf4, the transcriptional mediator of Wnt signalling (van Es et al, 2005). In addition, high Wnt activity prevents goblet cell differentiation, as goblet cells are absent in Apc-mutant intestine (Sansom et al, 2004). We have previously shown that Mapk signalling impedes the Wnt-induced maturation of Paneth cells and shifts the differentiation of common Paneth-goblet progenitors towards a goblet cell fate (Heuberger et al, 2014).

Besides the role of transcription factors, epigenetic regulation of gene expression has emerged as a powerful determinant of cell type identity as well as stem cell maintenance (Jadhav et al, 2016; Piunti & Shilatifard, 2016; Brand et al, 2019; Grinat et al, 2020; Goveas et al, 2021). We have recently reported a crucial role of the histone methyltransferase Mll1 in development of colorectal

---

[1]Cancer Research Program, Max Delbrück Center for Molecular Medicine (MDC) in the Helmholtz Society, Berlin, Germany    [2]Genomics, Center for Molecular and Cellular Bioengineering, Biotechnology Center, Technische Universität Dresden, Dresden, Germany    [3]DRESDEN-concept Genome Center, Center for Molecular and Cellular Bioengineering, Technische Universität Dresden, Dresden, Germany    [4]Medical Department, Division of Gastroenterology and Hepatology, Charité University Medicine, Berlin, Germany    [5]Berlin Institute for Medical Systems Biology, Max Delbrück Center for Molecular Medicine, Berlin, Germany    [6]Max Planck Institute of Molecular Cell Biology and Genetics, Dresden, Germany

Correspondence: julian.heuberger@charite.de
Johanna Grinat's present address is Epigenetics Programme, Babraham Institute, Cambridge, UK.

cancer (Grinat et al, 2020; Heuberger et al, 2021). Mll1 promotes a highly proliferative regenerative cell state which renders intestinal epithelial cells susceptible for tumorigenesis (Heuberger et al, 2021), maintains intestinal cancer stem cells, and promotes Wnt-induced tumorigenesis by antagonizing the polycomb repressive complex 2 (PRC2)–mediated repression of stem cell genes (Grinat et al, 2020). Ablation of Mll1 caused differentiation of Wnt-activated cancer stem cells by increasing a secretory gene expression profile (Grinat et al, 2020). Further work showed that Mll1 is required for intestinal stem cell maintenance at homeostasis (Goveas et al, 2021).

We here addressed the role of Mll1 in secretory cell fate determination in the adult intestinal epithelium at homeostasis. Using mouse genetics and intestinal organoid cultures, we show that Mll1 sustains the progenitor cell state and controls Wnt/Mapk–driven secretory cell specification into Paneth and goblet cells.

## Results

### Ablation of Mll1 causes aberrant secretory differentiation in intestinal epithelial crypts

To assess the role of Mll1 in intestinal cell fate determination, we ablated the expression of Mll1 in Lgr5+ stem cells by conditional mutagenesis. *Lgr5-EGFP-IRES-Cre^ERT2* mice, which express eGFP and a tamoxifen-inducible Cre recombinase under the control of the *Lgr5* promoter, were crossed with *Mll1^flox* mice to generate *Lgr5-EGFP-IRES-Cre^ERT2; Mll1^flox/+* and *Lgr5-EGFP-IRES-Cre^ERT2; Mll1^flox/flox* mice (referred to as Mll1+/− and Mll1−/−, respectively) (Barker et al, 2007; Denissov et al, 2014; Chen et al, 2019; Grinat et al, 2020). For lineage tracing, we further crossed in the *Rosa26-LacZ* reporter strain (Soriano, 1999). Mutagenesis was induced by intraperitoneal injections of tamoxifen. To test whether production of progeny from stem cells is affected by loss of Mll1, we first performed lineage tracing. Conditional mutagenesis using the *Lgr5-Cre^ERT2* caused a mosaic of recombined LacZ+ and adjacent non-recombined (wild-type) crypts (Fig S1A), as previously described (Schuijers et al, 2014; Grinat et al, 2020). We traced mutant cells post induction and observed that both Mll1+/− and Mll1−/− Lgr5+ stem cells gave rise to LacZ-positive progeny, which populated mutant crypts and villi starting from day 10 after induction of mutagenesis (Fig S1A). Ablation of Mll1 using the *Lgr5-Cre^ERT2* did not affect crypt cell proliferation, as seen through similar numbers of BrdU-incorporating cells in Mll1+/− and Mll1−/− crypts (Fig S1B, quantification on the right). Neither did Mll1-deficient cells undergo apoptosis, as shown by the absence of positive cleaved Caspase-3 staining in the crypts (Fig S1C). Hence, *Lgr5-Cre^ERT2*-driven ablation of Mll1 did not affect production of intestinal epithelial cells (Fig S1D, left panel). Heterozygous ablation of Mll1 (Mll1+/− mice) did not alter crypt cell morphology and cell type composition compared with wild-type control mice (Fig S1D, upper and middle right panel). Mll1−/− crypts, however, exhibited an increased number of cells with goblet cell-like features (Fig S1D, lower right panel, quantification below). By Alcian blue staining we identified these cells as mucus-containing goblet-like cells (Fig 1A, quantification on the right), as also previously described (Goveas et al, 2021). To exclude transient effects,

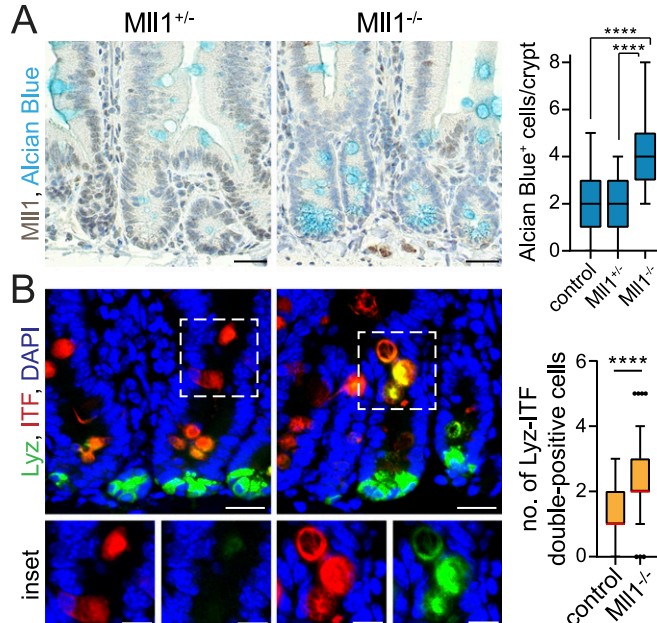

**Figure 1. Ablation of Mll1 causes aberrant secretory differentiation.**
**(A)** Representative immunohistochemistry for Mll1 and Alcian blue on sections of small intestinal crypts in Mll1+/− and Mll1−/− mice at 50 d after the induction of mutagenesis, nuclei counterstained with haematoxylin, scale bars 25 μm. Right: Quantification of Alcian blue–positive cells per Mll1+/−, Mll1−/− and adjacent non-recombined (control) crypts, counted from at least 15 crypts per mouse (Mll1+/− n = 2, Mll1−/− n = 4 independent mice), Mann–Whitney test, ****P < 0.0001. Box plot indicates median (middle line) and 25th, 75th percentile (box) with Tukey whiskers. **(B)** Representative immunostaining for Lyz (green) and ITF (red) on sections of Mll1+/− and Mll1−/− crypts at 50 d after induction of mutagenesis, nuclei in blue (DAPI), scale bars 20 μm. Magnifications of insets below, scale bars 10 μm. Right: Quantification of Lyz-ITF double-positive cells per crypt in Mll1+/− (control, n = 2 independent mice) and Mll1−/− intestines (n = 4 independent mice), counted from at least 20 crypts per mouse, Mann–Whitney U test, ****P < 0.0001. Box plot indicates median (red line) and 25th, 75th percentile (box) with Tukey whiskers.

we analysed mutant intestinal epithelia at 30–50 d after mutagenesis. Periodic acid-Schiff (PAS) staining of duodenum, jejunum, ileum, and colon epithelial sections revealed increases in mucus-producing goblet cells in all parts of Mll1-deficient intestinal epithelia with the exception of the ileum (Fig S1E). Increased goblet cell specification was most prominent in the jejunum. Using immunofluorescence, we observed an increased number of secretory cells in the upper parts of Mll1−/− jejunal crypts, which were double-positive for the goblet cell marker intestinal trefoil factor (ITF) (van der Sluis et al, 2006) and the Paneth cell marker lysozyme (Lyz) (Porter et al, 2002) (Fig 1B, quantification on the right). At the base of Mll1−/− crypts, Lyz-positive Paneth cells acquired Alcian blue–positive staining (compare Fig 1A with Figs 1B and S1F). In agreement with our previous characterisation of Mll1-deficient intestines (Goveas et al, 2021), the loss of Mll1 did not alter the number of chromogranin A (ChroA)-positive enteroendocrine cells per crypt-villus axis (Fig S1G, quantification on the right).

To explore the mechanistic role of Mll1 in secretory cell fate determination, we generated small intestinal organoids from *villinCre^ERT2; Mll1^flox/+* and *villinCre^ERT2; Mll1^flox/flox* mice (el Marjou et al, 2004; Denissov et al, 2014; Chen et al, 2019) and induced

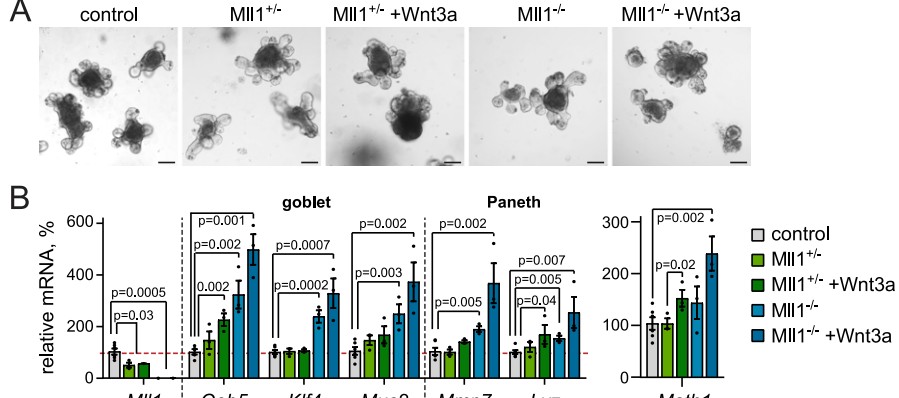

**Figure 2. Mll1-deficient organoids show aberrant Paneth/goblet cell differentiation.**
**(A)** Brightfield images of non-induced (control) and villinCre[ERT2]; Mll1[−/−] intestinal organoids at 5 d after 4-OHT-induced mutagenesis, scale bars 100 μm.
**(B)** mRNA expression of *Mll1* and secretory cell genes in non-induced control (grey), villinCre[ERT2]; Mll1[+/−] (green) and villinCre[ERT2]; Mll1[−/−] organoids (blue) at 7–12 d after 4-OHT-induced mutagenesis, n = 3 independent experiments, two-tailed unpaired *t* test. Data are presented as mean values ± SEM. Treatment with 500 ng/ml recombinant Wnt3a for 72 h.

mutagenesis in culture by addition of 4-OH tamoxifen. We tested for markers of Wnt activity and Paneth cell positioning (Batlle et al, 2002). Mll1 ablation neither changed the level of *Axin2* expression (Lustig et al, 2002; Grinat et al, 2020) (Fig S1H, right) nor did it alter the expression of *EphB2* and *EphB3* (Fig S1H, left), indicating that the diminished Paneth cell identity is not a result of decreased Wnt activity or an altered EphB2/3 gradient. Together, these findings suggest that the loss of Mll1 impairs Paneth/goblet cell specification and results in an accumulation of Paneth-goblet double-positive secretory cells, which appears to be caused by an intrinsic switch in cell fate rather than by decreased Wnt activity or an incorrect positioning of Paneth cells.

## Mll1-deficient intestinal organoids show increased expression of secretory Paneth and goblet cell genes

In the initial days after induction of mutagenesis, the loss of Mll1 did not alter organoid morphology (Fig 2A). However, RT–PCR analysis of Mll1-deficient organoids (Mll1[−/−]) revealed an up-regulation of the expression of the goblet cell–specific genes *Gob5* (Leverkoehne & Gruber, 2002), *Muc2* (van der Sluis et al, 2006) and the Krüppel-like factor *Klf4*, which is required for terminal goblet cell differentiation (Katz et al, 2002), as well as the Paneth cell–specific genes *Mmp7* and *Lyz* (Porter et al, 2002) (Fig 2B, light blue bars) compared with control and Mll1[+/−] organoids (grey and light green bars). In agreement, immunofluorescence stainings revealed that Mll1-deficient organoids exhibited increased numbers of ITF-positive goblet cells and Lyz-positive Paneth cells compared to non-induced control organoids (Fig S2A). To promote the production of Wnt-primed secretory progenitor cells, we treated the organoids with Wnt3a (Pinto et al, 2003; Heuberger et al, 2014), which resulted in increased expression of the secretory transcription factor *Math1* () (Fig 2B, right, dark green bar). Wnt3a stimulation of Mll1-deficient organoids caused enhanced *Math1* expression (Fig 2B, right, blue bars) and further increased the expression of the goblet cell genes *Gob5*, *Klf4*, *Muc2*, the Paneth cell genes *Mmp7* and *Lyz*, and the secretory genes *Gfi1* (Shroyer et al, 2005), *Spdef* (Gregorieff et al, 2009), and *Itf* (Mashimo et al, 1996) (Figs 2B and S2B, blue bars). The Notch-dependent transcription factor *Hes1*, which represses *Math1* induction (Yang et al, 2001; Fre et al, 2005), was not significantly

altered, indicating that Mll1 acts downstream of Notch-mediated repression to specify secretory cell differentiation (Fig S2C). The expression of the enteroendocrine-specific genes *Neurog3*, *chromogranin A* (*ChgA*), and *synaptophysin* (*Syp*) (Wiedenmann et al, 1988) as well as the absorptive enterocyte-specific *Fabp* and *Krt20* (Chan et al, 2009) was not increased upon loss of Mll1 (Fig S2D and E). The data suggest that Mll1 is involved in secretory cell fate decisions in intestinal epithelial crypts by controlling Paneth/goblet cell differentiation in Wnt-primed secretory progenitors.

## Mll1-deficient Wnt-high crypt cells exhibit goblet cell features

To consolidate our findings in vivo, we activated Wnt signalling by genetic stabilization of β-catenin through Cre-mediated deletion of the exon 3 of the β-catenin locus in *Lgr5-EGFP-IRES-Cre[ERT2]*; *β-catenin[deltaEx3/+]* mice (Harada et al, 1999; Barker et al, 2007) (hereafter called β-cat[GOF]) (Fig S3A, see nuclear location of β-catenin). High Wnt signalling in secretory progenitors induces differentiation into Paneth cells and prevents the maturation of goblet cells (Sansom et al, 2004; Andreu et al, 2005; van Es et al, 2005). Indeed, the high Wnt activity imposed a Paneth-like identity on the epithelial cells: β-cat[GOF]; Mll1[+/−] intestines showed high numbers of Mmp7- and Lyz-positive cells, and few ITF-positive goblet cells (Fig S3B). Alcian blue staining of control and β-cat[GOF] intestines confirmed reduced numbers of goblet cells in Wnt-high epithelium (Fig S3C). Ablation of Mll1 in β-cat[GOF]; Mll1[−/−] crypts caused accumulation of secretory cells double-positive for Paneth and goblet cell markers (Fig S3D), as it was observed in Mll1[−/−] crypts (see Fig 1). We isolated Paneth cells from β-cat[GOF]; Mll1[+/−] and β-cat[GOF]; Mll1[−/−] mice at 10 d after the induction of mutagenesis by fluorescence-activated cell sorting (FACS) and analysed transcriptomic changes by RNA sequencing. Immunohistochemistry (IHC) for Mll1 on intestinal crypts of β-cat[GOF]; Mll1[+/−] and β-cat[GOF]; Mll1[−/−] mice confirmed that all crypt cells including the long-lived Paneth cells at the crypt bottom were deficient for Mll1 at the time of sorting (Fig S3E). The transcriptome analysis revealed a differential regulation of numerous genes in the Paneth cells isolated from β-cat[GOF]; Mll1[−/−] mice (Fig S3F, left). Comparison with intestinal goblet and Paneth cell signatures (Haber et al, 2017) uncovered that β-cat[GOF]; Mll1[−/−] Paneth cells up-regulated the expression of several goblet cell–specific genes

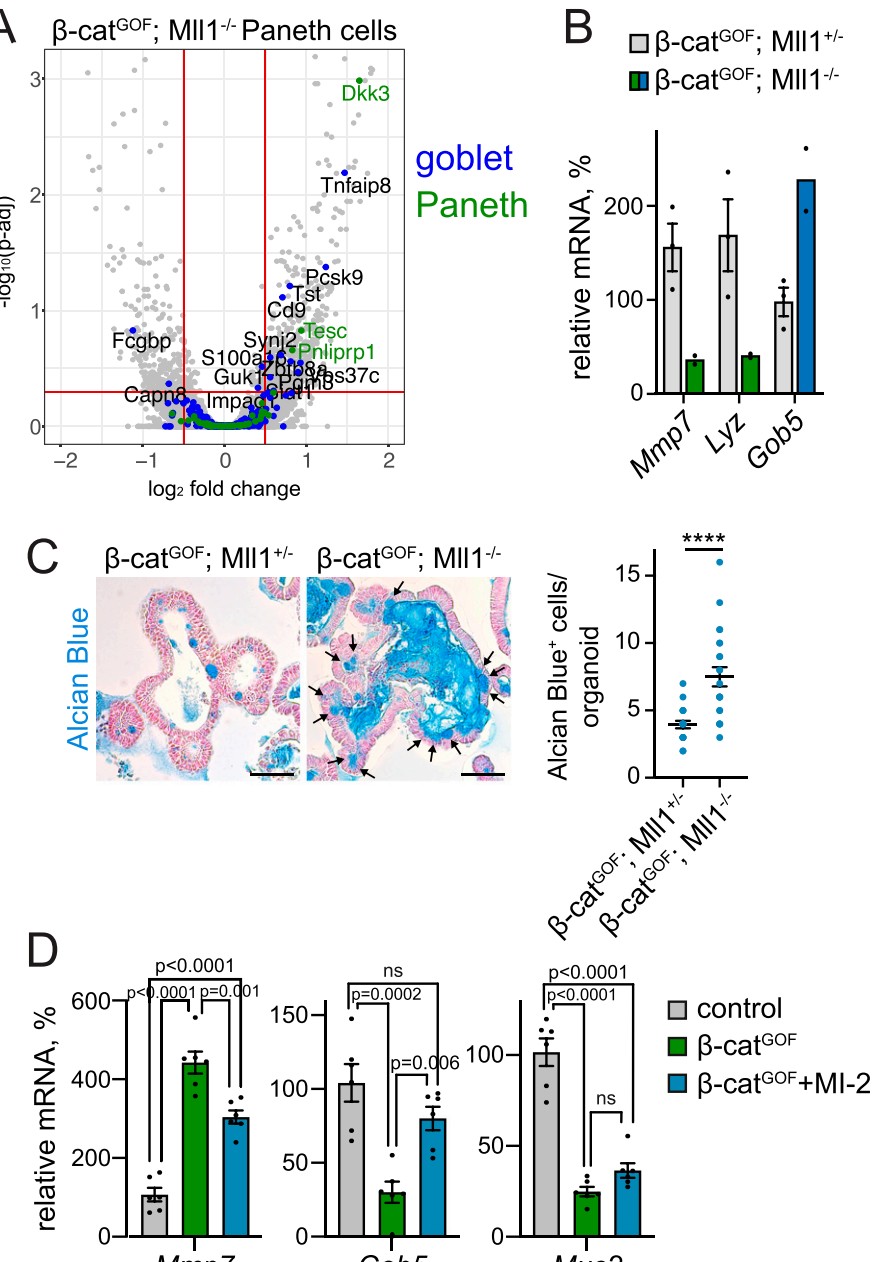

**Figure 3. Mll1-deficient Wnt-high Paneth cells and β-cat^GOF organoids exhibit goblet cell features.**
**(A)** Volcano plot of differentially expressed genes in Paneth cells isolated from β-cat^GOF; Mll1^+/− and β-cat^GOF; Mll1^−/− intestines at 10 d after induction of mutagenesis, n = 4 independent mice per genotype. Cut-offs log₂ fold-change ≥0.5, adjusted *P*-value (*P*-adj) ≤ 0.05. Goblet and Paneth cell–specific genes (from Haber et al [2017]) marked in green and blue, respectively. **(B)** mRNA expression of secretory genes (*Mmp7*, *Lyz*, and *Gob5*) in Paneth cells isolated from the intestines of n = 3 β-cat^GOF; Mll1^+/− and n = 2 β-cat^GOF; Mll1^−/− independent mice at 10 d post induction. Data are presented as mean values ± SEM. **(C)** Alcian blue stainings on sections of β-cat^GOF; Mll1^+/− and β-cat^GOF; Mll1^−/− organoids, nuclei counterstained with nuclear fast red, scale bars 50 μm. Black arrows indicate Alcian blue–positive goblet cells. On the right: Number of Alcian blue–positive cells in β-cat^GOF; Mll1^+/− and β-cat^GOF; Mll1^−/− organoids, at least 20 organoids quantified, two-tailed unpaired *t* test, ****P < 0.0001. Data are presented as scatter plot with mean ± SEM. **(D)** mRNA expression of Paneth cell gene *Mmp7* and goblet cell markers *Gob5* and *Muc2* in non-induced control, β-cat^GOF, and 48 h MI-2–treated β-cat^GOF organoids, n = 6 from three independent organoid lines, ordinary one-way ANOVA. Data presented as mean values ± SEM.

(Fig 3A). RT–PCR analysis of Paneth cells isolated from β-cat^GOF; Mll1^+/− and β-cat^GOF; Mll1^−/− mice at 10 d after the induction of mutagenesis showed a decrease in the expression of the Paneth cell genes *Mmp7* and *Lyz* upon loss of Mll1, whereas the expression of the early goblet-specific gene *Gob5* was increased (Fig 3B). The expression of the canonical Wnt target gene *Axin2* was not changed (Fig S3F).

We established β-cat^GOF intestinal organoids with heterozygous and homozygous deletion of Mll1 from *Lgr5-EGFP-IRES-Cre^ERT2*; β-cat^deltaEx3; *Mll1^flox* mice (Harada et al, 1999; Barker et al, 2007; Denissov et al, 2014; Chen et al, 2019) and induced mutagenesis in culture to obtain β-cat^GOF; Mll1^+/− and β-cat^GOF; Mll1^−/− organoids

(Fig S3G). As observed in vivo, the ablation of Mll1 in β-cat^GOF; Mll1^−/− organoids increased the expression of secretory genes specific for both Paneth (*Mmp7*) and goblet cells (*Klf4*, *Muc2*, and *Gob5*) (Fig S3H), substantiating enhanced secretory differentiation and a switch towards a goblet cell fate upon loss of Mll1. The Paneth cell marker *Lyz* was expressed in Mll1-deficient organoids, but its expression was not increased (Fig S3H). β-cat^GOF; Mll1^−/− organoids exhibited increased numbers of goblet cells and strong mucinous secretion towards the inside of the organoid, as assessed by Alcian blue staining (Fig 3C, quantification on the right). Inhibition of the Mll1 methyltransferase activity in β-cat^GOF organoids by MI-2 (Grembecka et al, 2012; Heuberger et al, 2021) reduced the

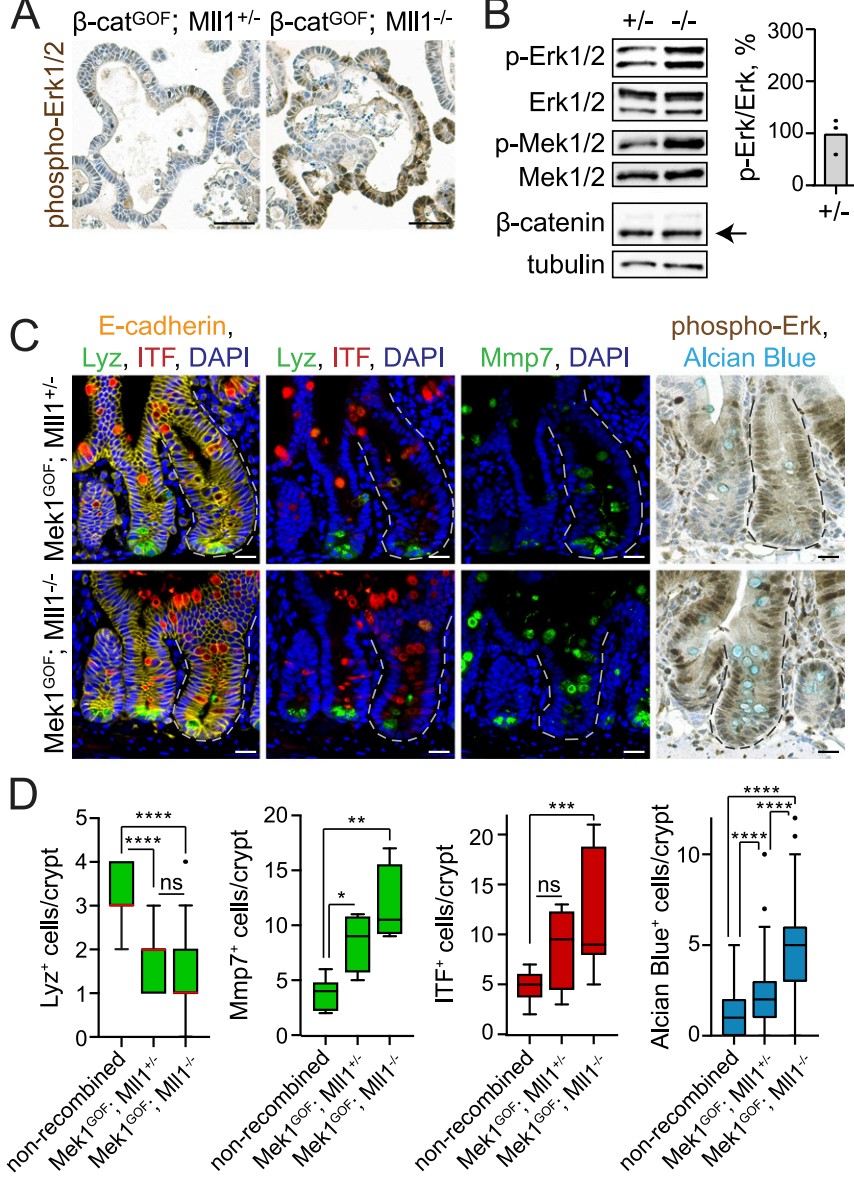

**Figure 4. Loss of Mll1 unleashes Mapk signalling and goblet cell differentiation in Wnt-high crypt cells and β-cat^GOF organoids.**

**(A)** Immunohistochemistry stainings for phospho-Erk1/2 on sections of β-cat^GOF; Mll1^+/− and β-cat^GOF; Mll1^−/− organoids, nuclei counterstained with haematoxylin, scale bars 50 μm. **(B)** Western blot for phospho-Erk1/2 and Erk1/2, phospho-Mek1/2 and Mek1/2, and β-catenin in β-cat^GOF; Mll1^+/− (+/−) and β-cat^GOF; Mll1^−/− (−/−) intestinal organoids. Right: Quantification of phospho-Erk1/2 relative to total Erk1/2 levels in β-cat^GOF; Mll1^+/− (+/−) and β-cat^GOF; Mll1^−/− (−/−) organoids, n = 2 independent organoid lines. Tubulin as loading control for β-catenin. **(C)** Left: immunofluorescence for Lyz (green) and ITF (red) on sections of Mek1^GOF; Mll1^+/− and Mek1^GOF; Mll1^−/− mouse small intestine at 10 d after induction. Recombined mutant crypts are surrounded by dashed lines. Middle: immunofluorescence staining for Mmp7 (green) on serial sections. E-cadherin (yellow) stains cell borders, nuclei in blue (DAPI), scale bars 20 μm. Right: immunohistochemistry for phospho-Erk1/2 and Alcian blue staining for goblet cells, nuclei counterstained with haematoxylin, scale bars 20 μm. Stainings were performed in five independent mice per genotype. **(D)** Number of Lyz-positive, Mmp7-positive, ITF-positive, and Alcian blue-positive cells in Mek1^GOF; Mll1^+/− and Mek1^GOF; Mll1^−/− crypts compared with adjacent non-recombined crypts, quantified from n = 2 independent mice per genotype (Alcian blue counted from at least 25 crypts per mouse, n = 5 independent mice each), Mann–Whitney U test, ****P < 0.0001, *P = 0.012, **P = 0.002, ***P = 0.0002. Box plot indicates median (red line) and 25^th, 75^th percentile (box) with Tukey whiskers.

expression of the Paneth cell gene *Mmp7* and reverted the Wnt-induced decrease in *Gob5* and by tendency *Muc2* expression (Fig 3D), suggesting that the function of Mll1 in secretory differentiation depends on its methyltransferase activity. Immunostaining revealed unchanged expression of β-catenin in β-cat^GOF; Mll1^+/− and β-cat^GOF; Mll1^−/− organoids (Fig S3I), revealing a role of Mll1 in fine-tuning Wnt-driven secretory specification independent of a global change in Wnt activity.

## Loss of Mll1 unleashes Mapk signalling in Wnt-activated crypt cells

In previous work we had shown that intestinal goblet cell differentiation depends on active Mapk signalling (Heuberger et al, 2014). We therefore investigated whether changes in Mapk signalling occur in Mll1-deficient and Wnt-activated crypt cells. IHC staining

on sections of β-cat^GOF; Mll1^+/− and β-cat^GOF; Mll1^−/− organoids revealed a strong increase in phospho-Erk1/2 levels upon loss of Mll1 (Fig 4A upper panel). Western blotting for phospho-Erk1/2 and phospho-Mek1/2 confirmed an increased activity of the Mapk pathway in β-cat^GOF; Mll1^−/− organoids compared with β-cat^GOF; Mll1^+/− organoids (Fig 4B, quantification on the right). Increased phospho-Erk1/2 levels were also detected in the mutant crypts of β-cat^GOF; Mll1^−/− mice at 10 d after induction, compared with β-cat^GOF; Mll1^+/− mice and adjacent non-recombined crypts (Fig S4A). This global increase in phospho-Erk1/2 levels was not detectable in the mutant crypts of Mll1^−/− mice with loss of Mll1 alone (Fig S4B, compare phospho-Erk1/2 levels in two Mll1-deficient crypts to non-recombined crypt on the right). Inhibition of Mapk signalling by treatment with the Mek inhibitor U0126 potently inhibited the growth of β-cat^GOF; Mll1^−/− organoids (Fig S4C). These data suggest that Mll1 prevents goblet cell differentiation of Wnt-activated crypt

cells by suppressing Mapk signalling, which is unleashed upon ablation of Mll1.

## Mll1 restricts Mapk-driven goblet cell differentiation

To further examine the role of Mll1 in the Wnt/Mapk-driven differentiation of secretory Paneth and goblet cells, we crossed in the *Mek1DD* allele (Cowley et al, 1994; Srinivasan et al, 2009). Cre-mediated recombination removes a transcription stop cassette and activates the expression of a constitutively active gain-of-function variant of *Mek1* (*Mek1DD*) from the *Rosa26* locus. We analysed *Mek1DD* mice with heterozygous and homozygous ablation of Mll1, *Lgr5-EGFP-IRES-Cre^ERT2*; *Mek1DD*; *Mll1^flox/+* and *Lgr5-EGFP-IRES-Cre^ERT2*; *Mek1DD*; *Mll1^flox/flox* mice (hereafter called Mek1^GOF; Mll1^+/− and Mek1^GOF; Mll1^−/−, respectively) at 10 d after the induction of mutagenesis. Activation of Mapk signalling by Mek1^GOF in Mek1^GOF; Mll1^+/− mice increased the proliferation of crypt cells, as seen through an increase in the incorporation of BrdU in mutant eGFP-positive crypts compared with adjacent non-recombined crypts (Fig S4D upper panel, quantification on the right). The eGFP is co-expressed with Mek1^GOF from an IRES-EGFP linked to the *Mek1DD* allele (Cowley et al, 1994; Srinivasan et al, 2009) and allows to distinguish recombined from non-recombined crypts. Of note, in *Lgr5-EGFP-IRES-Cre^ERT2*; *Mek1DD* mice the eGFP also identifies Lgr5-GFP⁺ stem cells (Barker et al, 2007). The homozygous ablation of Mll1 in Mek1^GOF; Mll1^−/− mice prevented the Mek1^GOF-induced hyper-proliferation and restricted proliferation to the base of the crypts (Fig S4D lower panel, quantification on the right). In accordance with the observed increase in cell proliferation, Mek1^GOF; Mll1^+/− intestines exhibited elongated villi and crypts (Fig S4E and F). The ablation of Mll1 reduced the Mek1^GOF-driven elongation of crypts and villi (Fig S4E and F). High Mapk signalling in the crypts decreased the number of Paneth cells at the crypt bottom, as shown by immunostaining for the Paneth cell marker Lyz (green) on serial sections of adjacent recombined and non-recombined crypts (Fig 4C upper left, quantification in Fig 4D). Mmp7 staining (green) revealed the presence of secretory Paneth-like cells further up in the crypts (Fig 4C upper middle, quantification in Fig 4D). Mek1^GOF; Mll1^+/− crypts did not show a strong increase in crypt-based goblet cells, as revealed by ITF (red) and Alcian blue staining compared with adjacent non-recombined crypts (Fig 4C upper panel). In contrast, Mek1^GOF; Mll1^−/− crypts were filled with secretory cells, which were positive for the Paneth cell marker Mmp7 (green) and exhibited a strong expression of goblet cell markers (ITF [red], Alcian blue) (Fig 4C lower panel, quantifications in Fig 4D). To note, loss of Mll1 did not impede the Mek1DD-induced activation of Mapk signalling, as shown by IHC for phospho-Erk1/2 in the mutant crypts of Mek1^GOF; Mll1^−/− mice compared with Mek1^GOF; Mll1^+/− mice (Fig 4C right panel). Furthermore, Mek1^GOF; Mll1^+/− and Mek1^GOF; Mll1^−/− crypts both exhibited nuclear β-catenin (Fig S4G, left panel) and in situ hybridisation revealed equal expression levels of the Wnt target gene *Axin2* (Fig S4G, right panel), demonstrating that Mek1 activation and loss of Mll1 did not change the global Wnt activity at the crypt base.

The data demonstrate that loss of Mll1 in the background of activated Mapk signalling has a dual effect, whereas loss of Mll1 prevents the Mek1DD-induced crypt hyper-proliferation and villus elongation, it on the other hand synergizes with Mapk signalling in

promoting generation of goblet cells. In other words, the presence of Mll1 promotes Mapk-induced progenitor cell proliferation and restricts goblet cell differentiation. Mll1-deficient secretory cells in Mek1^GOF; Mll1^−/− crypts were double-positive for Paneth and goblet cell markers and did not fully mature into goblet cells, as we had also observed in β-cat^GOF; Mll1^−/− crypts. These data indicate that Mll1 is critical for specifying Paneth and goblet cell fates. It safeguards the Wnt/Mapk-driven differentiation of secretory cells by preserving lineage-specific maturation.

## Mll1 controls the Wnt/Mapk-driven specification of Paneth and goblet cells

We established intestinal organoids from *Lgr5-EGFP-IRES-Cre^ERT2*; *β-cat^GOF*; *Mek1DD*; *Mll1^flox/+* and *Lgr5-EGFP-IRES-Cre^ERT2*; *β-cat^GOF*; *Mek1DD*; *Mll1^flox/flox* mice (hereafter called β-cat^GOF; Mek1^GOF; Mll1^+/− and β-cat^GOF; Mek1^GOF; Mll1^−/− mice, respectively) and induced mutagenesis in culture by in vitro administration of 4-OH tamoxifen. The β-cat^GOF; Mek1^GOF; Mll1^+/− organoids were highly proliferative, as revealed by staining for the proliferation marker Ki67 (Fig S5A, middle panel). The loss of Mll1 prevented the β-cat^GOF; Mek1^GOF-driven proliferation (Fig S5A, lower panel). We previously proposed opposing Wnt and Mapk signalling activities in goblet and Paneth cell lineage specification (Heuberger et al, 2014). In accordance, the simultaneous genetic activation of Wnt and Mapk signalling deregulated maturation of both Paneth (Lyz and Mmp7) and goblet cells (Alcian blue) (Fig 5A upper and middle panel, quantification in Fig 5B). These findings indicate the production of a proliferative non-specified cell population by simultaneous activation of Wnt and Mapk signalling. Remarkably, Paneth and goblet cells re-appeared in organoids with homozygous ablation of Mll1, largely as double-positive Paneth-goblet entities, as shown by immunostaining for Lyz, Mmp7, and ITF (Fig 5A lower panel, quantification in Fig 5B). RT–PCR analysis of β-cat^GOF; Mek1^GOF; Mll1^+/− and β-cat^GOF; Mek1^GOF; Mll1^−/− organoids confirmed the role of Mll1 in the control of Wnt/Mapk-driven secretory cell fate: The simultaneous activation of Wnt and Mapk signalling in the presence of Mll1 in β-cat^GOF; Mek1^GOF; Mll1^+/− organoids abrogated the expression of the Paneth cell genes *Mmp7* and *Lyz* (Fig 5C, dark green bars). The expression of the goblet cell gene *Klf4* was not decreased, which corresponded with the presence of ITF-positive goblet cells in β-cat^GOF; Mek1^GOF; Mll1^+/− organoids (Fig 5A middle panel, Fig 5C). Upon ablation of Mll1, the expression of *Lyz* was partly re-established and *Mmp7* expression strongly increased, as did the levels of the goblet cell–specific *Klf4* (Fig 5C, light green bars). These data from organoid culture support the role of Mll1 in controlling secretory cell specification induced by Wnt and Mapk.

To further corroborate the regulatory function of Mll1 downstream of Wnt and Mapk signalling we induced the production of stabilized β-catenin, Mek1^GOF and the loss of Mll1 in intestinal epithelia by intraperitoneal injections of tamoxifen in β-cat^GOF; Mek1^GOF; Mll1^+/− and β-cat^GOF; Mek1^GOF; Mll1^−/− mice. The intestines of β-cat^GOF; Mek1^GOF; Mll1^+/− mice became dysplastic and exhibited elongated villi (Fig S5B, quantification in Fig S5C). The mice did not survive beyond 20 d after the induction of mutagenesis. Homozygous ablation of Mll1 attenuated the β-cat^GOF; Mek1^GOF-induced hyperplastic phenotype. The villi were of normal (wild-type) length

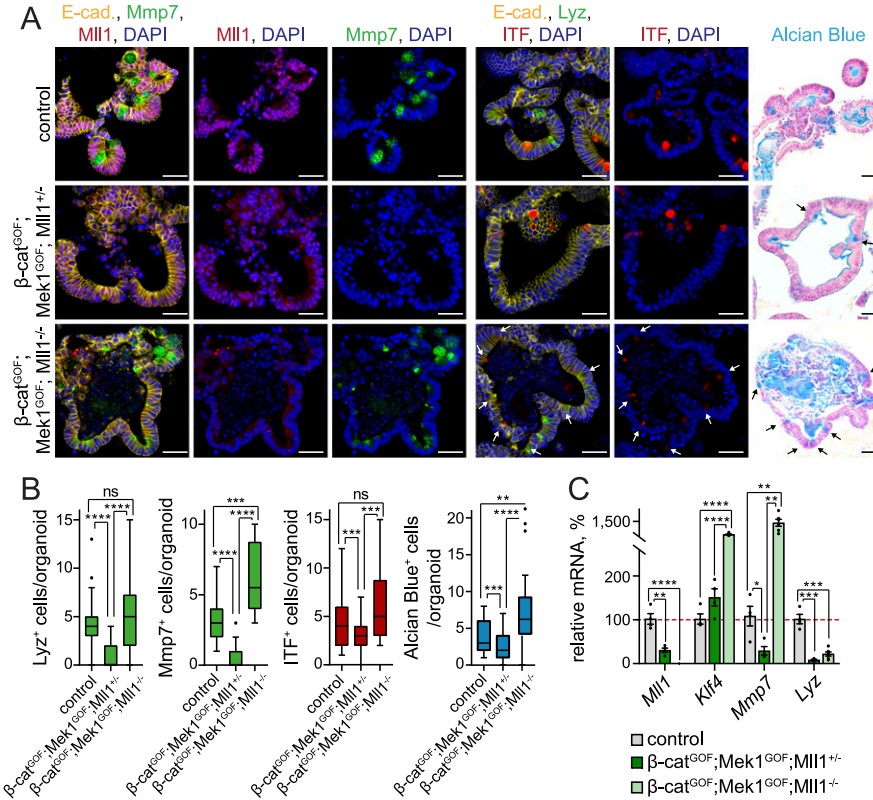

A

Figure 5. Loss of Mll1 induces mixed Paneth/goblet cell differentiation in intestinal organoids with concomitant activation of Wnt and Mapk signalling. (A) Left: immunofluorescence for Mmp7 (green) and Mll1 (red) on sections of non-induced (control) and 4-OHT–induced β-cat$^{GOF}$; Mek1$^{GOF}$; Mll1$^{+/-}$ and β-cat$^{GOF}$; Mek1$^{GOF}$; Mll1$^{-/-}$ intestinal organoids. Middle: immunofluorescence staining for Lyz (green) and ITF (red) on sections of organoids of the three genotypes, white arrows indicate cells with weak ITF expression. E-cadherin (yellow) stains cell borders and nuclei in blue (DAPI). Right: Alcian blue staining for goblet cells, nuclear fast red counterstaining, black arrows indicate Alcian blue–positive goblet cells. Scale bars 50 μm. (B) Quantifications of the numbers of Lyz-, Mmp7-, ITF-, and Alcian blue–positive cells per organoid in non-induced (control) and tamoxifen-induced β-cat$^{GOF}$; Mek1$^{GOF}$; Mll1$^{+/-}$; and β-cat$^{GOF}$; Mek1$^{GOF}$; Mll1$^{-/-}$ cultures, quantified from two independent organoid lines, Mann–Whitney U test, ****P < 0.0001, Mmp7 ***P = 0.0001, ITF ***P = 0.0006 (control-Mll1$^{+/-}$), ***P = 0.0002, Alcian blue ***P = 0.0008, **P = 0.001. Box plots indicate median (middle line) and 25$^{th}$, 75$^{th}$ percentile (box) with Tukey whiskers. (C) mRNA expression of Mll1 and the secretory Paneth/goblet cell genes Klf4, Mmp7, and Lyz in non-induced control and 4-OHT–induced β-cat$^{GOF}$; Mek1$^{GOF}$; Mll1$^{+/-}$ (n = 4) and β-cat$^{GOF}$; Mek1$^{GOF}$; Mll1$^{-/-}$ (n = 6) organoids, two-tailed unpaired t test, Mll1: ****P < 0.0001, **P = 0.0015, Klf4: ****P < 0.0001, Mmp7: *P = 0.017, **P = 0.004 (control-Mll1$^{-/-}$), **P = 0.003 (Mll1$^{+/-}$-Mll1$^{-/-}$), Lyz: ***P = 0.00013 (control-Mll1$^{+/-}$), ***P = 0.00015 (control-Mll1$^{-/-}$). Data are presented as mean values ± SEM.

(Fig S5B, quantification in Fig S5C). In the intestines of β-cat$^{GOF}$ mice, we observed an increased expression of the Paneth cell marker *Mmp7* and the secretory lineage marker *Spdef*, whereas the goblet cell marker *Gob5* strongly decreased (Fig S5D), reflecting Wnt-driven Paneth cell specification (van Es et al, 2005). Simultaneous activation of Wnt and Mapk signalling in β-cat$^{GOF}$; Mek1$^{GOF}$; Mll1$^{+/-}$ mice on the one hand decreased *Mmp7* expression and reverted the β-cat$^{GOF}$-induced expression of *Spdef* to control levels, and on the other hand reduced the expression of the goblet cell markers *Itf*, *Muc2*, and *Gob5* compared with wild-type control intestines (Fig S5D). The ablation of Mll1 in β-cat$^{GOF}$; Mek1$^{GOF}$; Mll1$^{-/-}$ mice re-established secretory cell differentiation in vivo, as demonstrated by up-regulation of *Spdef* and the Paneth cell–specific *Mmp7*, and by expression of the goblet cell markers *Itf*, *Muc2*, and *Gob5* (Fig S5D).

In summary, these data demonstrate that Mll1 keeps intestinal epithelial cells in an immature state while impeding secretory cell specification, and a decrease in Mll1 expression is necessary to allow for secretory differentiation. Our data further show that Mll1 also plays a role in specifying the Wnt/Mapk–driven differentiation of Paneth and goblet cells. In Wnt-high crypt cells, Mll1 suppresses a Mapk-induced goblet cell fate and thereby promotes Paneth cell specification. The loss of Mll1 unleashes Mapk signalling and cell-intrinsically perturbs segregation of secretory Paneth and goblet cell fates.

### Mll1 regulates Gata4/6 expression to restrict Mapk signalling and goblet cell fate

We previously reported that Mll1 regulates the expression of the transcription factor Gata4 to sustain stemness and restrict secretory goblet cell differentiation of β-cat$^{GOF}$ intestinal cancer stem cells (Grinat et al, 2020). Mll1-deficient crypts exhibited a decreased expression of Gata4 (Fig S6A). *Gata4* expression was reduced in villinCre$^{ERT2}$; Mll1$^{-/-}$ organoids compared with non-induced control and villinCre$^{ERT2}$; Mll1$^{+/-}$ organoids (Fig S6B). As shown in Fig 2, the Mll1$^{-/-}$ organoids did not exhibit altered *Math1* expression unless stimulated with Wnt3a (see Fig 2B). In accordance, β-cat$^{GOF}$; Mll1$^{-/-}$ organoids showed a decreased expression of *Gata4*, compared with β-cat$^{GOF}$ organoids (Fig 6A). In these Wnt-high Mll1-deficient organoids, the decreased *Gata4* expression was associated with an up-regulation of the secretory transcription factor *Math1* (Fig 6B), matching previous reports that ablation of Gata4 enhances intestinal *Math1* levels and causes aberrant goblet cell differentiation (Beuling et al, 2011; Kohlnhofer et al, 2016). We established an inducible CRISPR/Cas9–mediated knockout system in wild-type intestinal organoids to ablate the expression of Gata4. IHC stainings for Gata4 and phospho-Erk1/2 on serial sections of sgGata4 organoids at 8 d after induction of Cas9 and mutagenesis revealed increased phospho-Erk1/2 levels and Alcian blue-positive goblet cells in Gata4-negative areas (Fig 6C). These data suggest that the increased Mapk activity and secretory goblet cell differentiation of Mll1-deficient cells is due to decreased levels of Gata4.

To further define the role of Mll1 in the regulation of Gata4 and secretory cell differentiation, we studied the Wnt-high human colon cancer cells Ls174T and DLD1 (van de Wetering et al, 2002; van der Flier et al, 2007), from which we had previously established MLL1 knockdown lines (Grinat et al, 2020). We used the CRISPR/Cas9 technology to ablate the expression of GATA6, the colon homologue of Gata4 (Whissell et al, 2014), in Ls174T cells (Fig S6C). As observed

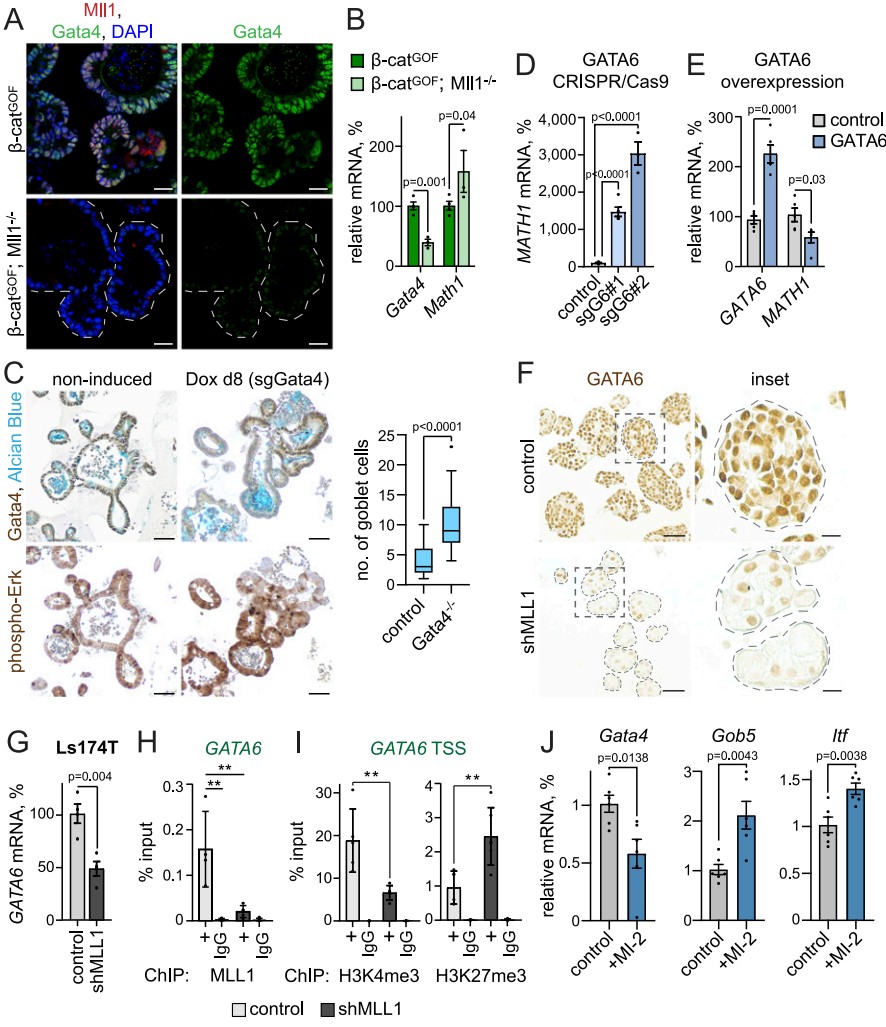

**Figure 6.    Mll1 controls Mapk signalling and goblet cell fate through regulation of Gata4/6 expression.**
**(A)** Immunofluorescence for Mll1 (red) and Gata4 (green) on sections of 4-OHT–induced β-cat$^{GOF}$ and β-cat$^{GOF}$; Mll1$^{-/-}$ organoids. Nuclei in blue (DAPI), scale bars 20 $\mu$m. **(B)** mRNA expression of *Gata*4 and *Math1* in β-cat$^{GOF}$ and β-cat$^{GOF}$; Mll1$^{-/-}$ organoids, n = 3 independent experiments, two-tailed unpaired *t* test. Data are presented as mean values ± SEM. **(C)** Immunohistochemistry for Gata4 (upper panel) and phospho-Erk (lower panel) on serial sections of non-induced control and doxycycline (Dox)-induced sgGata4 organoids at day 8 after Cas9 induction, nuclei counterstained with haematoxylin. Alcian blue stains mucus and goblet cells, scale bars 50 $\mu$m. Right: quantification of the number of goblet cells in non-induced control organoids and Gata4-negative areas of sgGata4 organoids, counted in at least five organoids from two sgGata4 organoid lines in three independent rounds of Cas9 induction, Mann–Whitney *U* test. Box plot indicates median (middle line) and 25$^{th}$, 75$^{th}$ percentile (box) with Tukey whiskers. **(D)** mRNA expression of *MATH1* in control and two independent sgGATA6 Ls174T cell clones, n = 3 experiments, two-tailed unpaired *t* test. Data are presented as mean values ± SEM. **(E)** mRNA expression of *GATA6* and *MATH1* in Ls174T cells at 72 h after transfection with GATA6 cDNA (blue bars) compared to control cells (grey bars), n = 5 replicates from four independent experiments, two-tailed unpaired *t* test. Data are presented as mean values ± SEM. **(F)** Immunohistochemistry for GATA6 on sections of control and shMLL1 Ls174T spheres, scale bar 50 $\mu$m, inset 25 $\mu$m. **(G)** mRNA expression of *GATA6* in control and shMLL1 Ls174T sphere cells, n = 4 independent experiments with three biologically independent samples, two-tailed unpaired *t* test. Data are presented as mean values ± SEM. **(H)** ChIP for MLL1 in control (light grey columns) and 6d doxycycline-induced shMLL1 Ls174T cells (black columns), binding at the *GATA6* promoter, represented as % input, n = 5 replicates derived from two biologically independent cell clones over four independent experiments, two-tailed unpaired *t* test, significance calculated for control versus shMLL1 and IgG, **P = 0.002, **P = 0.007. Data are presented as mean values ± SD. **(I)** ChIP for H3K4me3 and H3K27me3 at the transcriptional start site of *GATA6* in control (grey columns) and 11d doxycycline-induced shMLL1 Ls174T cells (black columns), represented as % input, n = 5 replicates from two biologically independent cell clones over four independent experiments, two-tailed unpaired *t* test, significance calculated for control versus shMLL1, **P = 0.007, **P = 0.004. Data are presented as mean values ± SD. **(J)** mRNA expression of *Gata*4 and goblet cell markers *Gob5* and *Itf* in wild-type control organoids and upon 48 h of Mll1 inhibition with 1.5 $\mu$M MI-2, n = 6 from three independent organoid lines, two-tailed unpaired *t* test. Data are presented as mean values ± SEM.

in β-cat$^{GOF}$; Mll1$^{-/-}$ organoids, CRISPR/Cas9-mediated knockout of GATA6 in Ls174T cells increased the expression of the secretory transcription factor *MATH1* (Fig 6D). Vice versa, transient over-expression of GATA6 in Ls174T cells decreased *MATH1* expression (Fig 6E). IHC staining for GATA6 on control and shMLL1 Ls174T spheres revealed a strong decrease in GATA6 levels upon depletion of MLL1 (Fig 6F). MLL1 knockdown spheres with decreased expression of GATA6 exhibited increased levels of LYZ and Alcian blue staining (Fig S6D and E). We performed chromatin immunoprecipitation (ChIP) of MLL1 in Ls174T and DLD1 colon cancer spheres, in which knockdown of MLL1 decreased the mRNA expression of *GATA6* (Figs 6G and S6F). ChIP revealed MLL1 binding to the *GATA6* promoter (Figs 6H and S6G and H). The enrichment was reduced in shMLL1 Ls174T and DLD1 cells, and was absent at a negative control region 70 kb downstream of the *TAL1* promoter. Activating H3K4me3 at the *GATA6* transcriptional start site switched to repressive H3K27me3 upon knockdown of MLL1 (Fig 6I). To corroborate the

enzymatic function of Mll1 in controlling secretory goblet cell differentiation we treated wild-type intestinal organoids with MI-2. After 48 h of MI-2 treatment, the organoids showed decreased expression of *Gata*4 and increased expression of the goblet cell markers *Itf* and *Gob5* (Fig 6J). These data reveal the Mll1 methyltransferase as an upstream regulator of a Gata4/6-Math1 axis that counters Mapk activity and goblet cell differentiation in Wnt-high secretory progenitor cells.

# Discussion

Here we report a role of the epigenetic regulator Mll1 in Wnt- and Mapk-driven cell fate specification of secretory progenitors in the intestinal epithelium. We show that Mll1 promotes proliferation and the progenitor cell state, and impedes secretory differentiation into goblet and Paneth cells. Our data reveal an Mll1-dependent regulation of the dual role of Mapk signalling in progenitor cells of the

TA zone, promoting cell proliferation in Mll1-high TA cells and instructing goblet cell differentiation in Mll1-low progenitor cells. Ablation of Mll1 resulted in cells double-positive for Paneth and goblet cell markers, revealing a role of residual Mll1 in segregating Wnt- and Mapk-instructed Paneth and goblet cell fates. Our data provide evidence for a dual role of Mll1 in progenitor cell maintenance and cell fate determination in the intestinal epithelium.

Cell fate choice in the TA zone is guided by an interplay of various signalling pathways. Notch signalling controls the specification of absorptive versus secretory lineages. Inhibition of Notch signalling up-regulates the expression of *Math1* and causes accumulation of secretory cells double-positive for the Paneth and goblet cell markers Mmp7 and Muc2 (van Dussen et al, 2012). The differentiation of the secretory Paneth and goblet cells critically depends on Wnt and Mapk activities, respectively (Heuberger et al, 2014). We observed that simultaneous activation of Wnt/$\beta$-catenin and Mek1/Mapk signalling abrogated secretory cell maturation in intestinal organoids and imposed an immature precursor state, as we had previously suggested (Heuberger et al, 2014). Whereas $\beta$-cat$^{GOF}$-induced Wnt activation promoted a Paneth cell fate and prevented Mapk-induced goblet cell differentiation, the ablation of Mll1 in $\beta$-cat$^{GOF}$ and in $\beta$-cat$^{GOF}$; Mek1$^{GOF}$ cells led to re-appearance of goblet cells, largely as mixed Paneth-goblet entities. This suggests that Mll1 restricts Mapk signalling and goblet cell specification of secretory progenitor cells. Of note, the loss of Mll1 results in increased goblet cell specification in the wild-type intestine but a global increase in Mapk activity is not visible, which we ascribe to the transient nature and small population of secretory progenitor cells, in which this regulatory circuit is active. Consistent with the role of Wnt signalling for Math1-positive secretory progenitors (Tian et al, 2015), Wnt activation by Wnt3a treatment or $\beta$-cat$^{GOF}$ expands the secretory progenitor pool and hence allows to detect molecular and signalling effects. In Mll1-competent Mek1$^{GOF}$ crypts, Mapk signalling promoted cell proliferation. The ablation of Mll1 shifted the effect of Mapk signalling from pro-proliferative to induction of goblet cell differentiation. Mek1$^{GOF}$; Mll1$^{-/-}$ crypts exhibited increased numbers of goblet cells. The presence of Mll1 thus maintains a proliferative progenitor state and restricts goblet cell differentiation. This closely fits our observation of high levels of Mll1 in TA cells, in which active Mapk signalling promotes cell proliferation (Heuberger et al, 2014; Grinat et al, 2020).

The accumulating secretory cells in Mek1$^{GOF}$; Mll1$^{-/-}$ crypts were double-positive for Paneth and goblet cell markers despite high Mapk activity and absence of Mll1, indicating that Mll1 is essential for preserving the lineage specification of Paneth and goblet cells. Like Mll1, polycomb PRC2 complexes have been implicated in the maintenance of intestinal stem and progenitor cells: their genetic ablation causes loss of stem cells and aberrant secretory differentiation (Chiacchiera et al, 2016; Jadhav et al, 2016). Deletion of PRC2 has been shown to disrupt the intestinal differentiation program and cause accumulation of mixed Paneth/goblet cells in intestinal crypts (Koppens et al, 2016). In the light of our data, this highlights the importance of a proper balance of Mll1 and PcG activities for establishing and maintaining cell identity in Paneth/goblet cell progenitors. Our data suggest that Mll1—like PRC2—exerts a dual role in intestinal homeostasis: it sustains intestinal progenitor cells and needs to be down-regulated to enable terminal differentiation (Grinat et al, 2020; Goveas et al, 2021), but also plays a role in controlling cell fate specification of secretory Paneth and goblet cells.

The transcription factors Gata4 and Gata6 are required for secretory cell differentiation and lineage maturation in the epithelium of the small intestine and colon (Beuling et al, 2011). Gata4 is expressed throughout the small intestinal epithelium of the jejunum, but is absent in goblet and enteroendocrine cells (Bosse et al, 2006). In line with our data, the ablation of Gata4 promotes maturation of goblet cells (Beuling et al, 2011). However, we did not observe a strong effect of Mll1 and Gata4 depletion on the enteroendocrine cell fate. Mll1$^{-/-}$ organoids showed a slightly reduced expression of the enteroendocrine progenitor marker *Neurog3*, which may further promote the acquisition of a goblet cell fate (Li et al, 2021), but the number of differentiated enteroendocrine cells was unchanged in Mll1$^{-/-}$ intestine.

A recent study reported a negative regulation of Mapk signalling by Gata4 in the developing stomach epithelium through Gata4-mediated expression of negative regulators such as *Spry2*, *Dusp4*, and *Dusp6* (Sankoda et al, 2021). In the intestine, loss of Dusp6, which would correspond to a loss of Mll1/Gata4, results in increased numbers of goblet cells without affecting enteroendocrine cells (Beaudry et al, 2019). This might also happen in the TA cells of the intestinal epithelium, where Mapk activity must be balanced between promoting cell proliferation and inducing goblet cell differentiation. Loss of negative Mapk regulator expression in Gata4-low cells increases Mapk activity and promotes goblet cell differentiation. Our data reveal Mll1 as an epigenetic regulatory hub in the crosstalk of Wnt and Mapk signalling in the context of secretory cell specification. By sustaining the expression of Gata4/6 transcription factors, Mll1 balances opposing Wnt and Mapk activities to suppress goblet cell specification and promote the alternate Paneth cell fate in mixed-lineage secretory progenitors. A similar mechanism has been described in the immune system, where Mll1 regulates Gata3 to specify and maintain memory Th2 cells (Yamashita et al, 2006). In intestinal development, a suppressive function of Mapk signalling on Wnt signalling has been reported (Wei et al, 2020). In lineage specification in the adult intestinal epithelium, Mapk signalling suppresses Wnt-driven Paneth cell specification (Heuberger et al, 2014). Crosstalk of Wnt and Mapk signalling is also frequently observed in cancer, where both pathways regulate each other either negatively or positively dependent on the cancer type (Guardavaccaro & Clevers, 2012). In oncogenic Wnt-activated intestinal cancer cells, the loss of Mll1 promotes the Mapk-dependent goblet cell fate (Grinat et al, 2020). We here show that loss of Mll1 in Mapk-activated cells results in the production of mixed Paneth/goblet cells. Concomitant genetic activation of Wnt and Mek1 results in non-specified cell populations that differentiate upon ablation of Mll1. Thus, Mll1 emerges as a regulatory module of Wnt and Mapk signalling.

Altogether, our study illustrates the interplay of epigenetic and signalling cues in the control of cell fate specification in adult tissues. Our data unravel Mll1 as an epigenetic factor in the regulation of secretory lineage specification in the intestinal epithelium. Mll1 coordinates Wnt and Mapk signalling to sustain progenitor cell proliferation and specify secretory Paneth and goblet cell fates through Gata4/6 transcription factors.

# Materials and Methods

## Mice

Mice were bred in pathogen-free conditions, and care and use of animals were performed according to the European and national regulations, published in the Official Journal of the European Union L 276/33, 22 September 2010. Transgenic mouse lines used have been previously described: Lgr5-EGFP-IRES-Cre^ERT2 (Barker et al, 2007), villin-Cre^ERT2 (el Marjou et al, 2004), Mll1^flox (Denissov et al, 2014; Chen et al, 2019), Rosa26-lacZ (Soriano, 1999), $\beta$-cat^GOF (Harada et al, 1999), and Mek1DD (Cowley et al, 1994; Srinivasan et al, 2009). Mutagenesis was induced in 4–6-wk-old mice by intraperitoneal injections of tamoxifen (50 mg/kg; Sigma-Aldrich, diluted 1:10 in sunflower oil) on three consecutive days. Mice were analysed at the indicated time points after the last tamoxifen injection. Mice were given i.p. injections of BrdU (Millipore), final concentration 50 $\mu$g/g of body weight in PBS at 2 h before euthanasia. Both females and males were analysed.

## Isolation of Paneth cells and RNA sequencing

*Lgr5-EGFP-IRES-Cre^ERT2; Mll1^flox/+* and *Lgr5-EGFP-IRES-Cre^ERT2; Mll1^flox/flox* littermates (n = 4) were given tamoxifen via i.p. injections for three consecutive days and were dissected 10 d later. For isolation of Paneth cells, crypts were dissociated into single cells with TrypLE Express (Thermo Fisher Scientific) for 30 min at 37°C. Dissociated cells were passed through 70 $\mu$m cell strainer and washed wit 5% FBS/PBS. Cells were stained with PE-conjugated anti-CD24 (1:100 dilution, Cat. no. 12-0242-81; eBioscience) antibody, APC-conjugated anti-326 epithelial cell adhesion molecule (EpCAM) antibody (1:100 dilution, Cat. no. 17-5791-80; eBioscience) and Alexa-Fluor 700 CD45 antibody (1:50 dilution, 560693; BD) for 45 min on ice. Stained cells were collected by centrifugation and resuspended in SYTOX blue dead cell stain (1:20,000 dilution; Thermo Fisher Scientific). FACS sorting for viable EpCAM^+ Cd24^+ Paneth cells was performed on a FACS AriaTM III cell sorter (BD). 300 single cells were sorted into a PCR tube containing 2 $\mu$l of nuclease-free H$_2$O with 0.2% Triton-X 100 and 4 U murine RNase Inhibitor (New England Biolabs) and stored at –80°C. RNA isolation and library preparation were performed based on the Smart-seq2 protocol (Picelli et al, 2013). In brief, RNA was denatured for 3 min at 72°C in the presence of 2.4 mM dNTP (Invitrogen), 240 nM dT-primer, and 4 U RNase Inhibitor (New England Biolabs) and reverse-transcribed using the Superscript II Reverse Transcriptase (Invitrogen). Single-stranded cDNA was amplified with the Kapa HiFi HotStart Readymix (Roche) and purified using Sera-Mag SpeedBeads (GE Healthcare). cDNA quality and concentration were determined with the Fragment Analyzer (Agilent Technologies). Sequencing was performed on a Nextseq500 (Illumina) with a sample sequencing depth of 30mio reads on average. RNA seq reads were aligned to the mm10 transcriptome with GSNAP (version 2018-07-04) and a table of read counts per gene was created based on the overlap of the uniquely mapped reads with the Ensembl Gene annotation (version 92), using featureCounts (version 1.6.3). Read counts were further processed with the DESeq2 R package (version 1.22.2). Sample-to-sample correlation was computed by Euclidean distance between samples based on the normalized counts. Differential

gene expression analysis was performed with DESeq2, for which a maximum of 10% false discovery rate (FDR) was accepted. Volcano plots were generated by a generic R X-Y plotting using the log$_2$-FoldChange versus the log$_2$ adjusted *P*-values. RNA seq differential expression data can be found in Supplemental Data 1. RNA seq data are available under GEO accession number GSE177047.

## Organoid culture

Intestinal organoids were obtained as previously described (Heuberger et al, 2014; Grinat et al, 2020): small intestines of mice were dissected and dissociated in 8 mM EDTA/PBS for 5 min at RT followed by 20 min incubation in ice-cold 2 mM EDTA/PBS at 4°C. The epithelia were fractionated by shaking in ice-cold PBS. 250 crypts were seeded in 20 $\mu$l of growth factor-reduced Matrigel (Matrigel 356231; BD) and cultured in basic crypt medium (60/40 Advanced DMEM/F12 supplemented with N2 and B27, GlutaMax, N-Acetylcysteine, and Penicillin/Streptomycin [Invitrogen]) containing 50 ng/ml mEGF (Gibco), 100 ng/ml mNoggin (PeproTech) and 500 ng/ml hR-Spondin1 (PeproTech) (ENR medium). Organoids were split every 4–6 d by mechanical disruption. Cre-mediated recombination was induced by addition of 800 nM 4-hydroxy-tamoxifen (4-OHT) for 2 d. $\beta$-cat^GOF organoids were selected by R-Spondin1 withdrawal. All analyses were performed at 7–10 d after mutagenesis. Recombinant Wnt3a (1324-WN; R&D Systems) was added for 72 h. 5 $\mu$M U0126 (Cat. no. 662005; Calbiochem) treatment for 72 h in EGF-free crypt medium supplemented with Noggin and R-spondin1, 1.5 $\mu$M MI-2 (Cat. no S7618; Selleckchem) treatment for 48 h.

## Cell culture

DLD1 and Ls174T human colon cancer cell lines were cultured in 1× DMEM supplemented with 10% FBS and 1% Penicillin/Streptomycin (PenStrep) at 37°C and 5% CO$_2$. Cell line identity was confirmed by Multiplex human Cell line Authentication (Multiplexion). Inducible pInd11-shMLL1 knockdown cell lines were generated as previously described (Meerbrey et al, 2011; Fellmann et al, 2013; Grinat et al, 2020). For GATA6 overexpression, Ls174T cells were transiently transfected with pcDNA3.1-GATA6 (clone OHu27933; GenScript) or pcDNA3.1 empty vector control using Lipofectamine 2000 (Invitrogen) and harvested for RNA isolation at 72 h after transfection.

## CRISPR/Cas9 genome editing

### Generation of lentiviral particles
For CRISPR/Cas9-mediated genome editing, the TLCV2 vector system was used (Addgene plasmid #87360, a gift from Adam Karpf). sgRNAs targeting mouse Gata4 (sgGata4-#1: 5′-GTCATCAAACA-TATCTACTG-3′ and sgGata4-#3: 5′-AGAACCATGTCTGCTCTGGT-3′; sgGata4-#4fwd: 5′-GACAGACTGATCTATAATCG-3′ and sgGata4-#4rev: 5′-GTGTGTGTGATGGAATGTAG-3′) or human GATA6 (sgGATA6#1: 5′-TTTCTAGCCTTCATCACGG-3′, sgGATA6#2: 5′-GCAATCATCTGAGTTAGAAG-3′) were individually cloned into TLCV2. For production of lentiviral particles, 293TN cells were co-transfected with 10 $\mu$g psPAX2, 2.5 $\mu$g pMD2.G, and 10 $\mu$g TLCV2 containing the respective sgRNA by transfection with PEI (Sigma-Aldrich). Lentiviral particle–containing supernatants were collected at 24 and 48 h post transfection, passed

through a 0.45-μm filter and concentrated with Lenti-X Concentrator (Takara).

### Transduction of intestinal organoids

Intestinal organoids were cultured in ENR medium supplemented with 3 μM CHIR99021 and 500 ng/ml Wnt3a for 96 h before lentiviral transduction. Organoids were dissociated with TrypLE (Gibco) for 2 min in a 37°C water bath, washed once with 0.1% BSA/PBS, resuspended in ENR medium supplemented with 500 ng/ml Wnt3a, 10 μM Y27632 (#A9165; Sigma-Aldrich), and Transdux (#631231; System Bioscience) and mixed 1:1 with the lentiviral particles (1:1 mix of TLCV2-sgGata4-#1 and -#3 and TLCV2-sgGata4-#4fwd and -#4rev, respectively). Spinoculation was performed in PolyHEMA-coated 48-well plates at 1,000g for 1 h at RT, followed by 5-h incubation at 37°C. Transduced cells were then washed with 0.1% BSA/PBS, seeded in Matrigel, and cultured in ENR medium supplemented with 500 ng/ml Wnt3a and 10 μM Y27632 for 2 d. The medium was changed to basic ENR crypt medium and transduced organoids were selected with 1 μg/ml puromycin for 3 d starting from day 4 after transduction. Cas9 activity in stable organoids was induced with 600 ng/ml doxycycline (D5897; LKT Laboratories) for 3 d. Gata4 knockout organoids were analysed 5 d after mutagenesis.

### Transduction of Ls174T cells

For lentiviral transduction, lentiviral particles (TLCV2-sgGATA6#1 or TLCV2-sgGATA6#2) were mixed 1:1 with fresh growth medium containing 8 μg/ml polybrene. Ls174T cells were spinoculated at 300g for 1 h at RT, followed by overnight incubation at 37°C. Transduced cells were selected with 1 μg/ml puromycin for 3 d starting from day 4 after transduction. Cas9 activity was induced with 600 ng/ml doxycycline (D5897; LKT Laboratories) for 3 d. Induced cells were sorted by FACS for GFP, which is co-expressed with Cas9 from TLCV2 and cultured as single cell clones to establish Ls174T sgGATA6 cell lines. One representative cell clone per sgRNA was selected for further analysis and compared with non-induced parental control.

### Histology, immunofluorescence staining, and analysis

Murine tissue was fixed in 4% formaldehyde/PBS, and histological analyses were performed on 5–7-μm paraffin sections. Organoids were fixed in 4% formaldehyde/PBS for 1 h and embedded in 1.5% agarose/PBS. For immunostaining on paraffin sections, antigen retrieval was carried out by boiling in 10 mM sodium citrate, pH 6.0, or 1 mM EDTA 20 mM Tris, pH 8.5. For IHC, sections were incubated in 4% $H_2O_2$/PBS for 5 min before 1-h incubation with blocking solution (0.1% Tween 20, 5–10% horse serum, 1% BSA in PBS) followed by incubation of the primary antibody diluted in blocking solution overnight at 4°C. For Mll1 staining, antigen retrieval was performed by boiling in 10 mM sodium citrate, pH 6.0, for 15 min, and sections were permeabilized in ice-cold methanol for 10 min before incubation in blocking solution. Fluorochrome-conjugated or HRP-coupled secondary antibodies and DAPI were incubated in blocking solution for 1–2 h at RT. IHC was developed with the DAB chromogenic substrate (DAKO), dehydrated and mounted with non-aqueous mounting medium (Entellan). Fluorochrome-conjugated

or HRP-coupled secondary antibodies were incubated for 1 h at RT. IHC was developed with the DAB chromogenic substrate (DAKO). For H&E staining, tissue sections were incubated with haematoxylin solution for 1 min and stained with eosin for 5 min before dehydration and mounting with non-aqueous mounting medium. Alcian blue solution (pH 2.5 in 3% acetic acid) was incubated for 30 min, and sections were counterstained with nuclear fast red for 5 min or haematoxylin for 30 s. PAS staining was performed using the PAS staining kit (#101646; Millipore).

### In situ hybridization

5-μm-thick intestinal paraffin sections were rehydrated and treated with 0.2 N hydrochloric acid and proteinase K. Slides were then post-fixed with 4% formaldehyde/PBS, demethylated with acetic anhydride, and prehybridized. Hybridization with RNA probes was performed in a humid chamber with 1 μg/ml digoxigenin-labelled *Axin2* RNA probe for 24 h at 63°C (Klaus et al, 2007). Slides were washed, blocked, and incubated with anti-digoxigenin-alkaline phosphatase conjugate overnight at 4°C. The stainings were developed with BM purple and counterstained with Pyronin G.

### Light microscopy and data analysis

Representative z-stacks were acquired with inverted laser scanning microscopes LSM710 and LSM700 using 405, 488, 561, and 633-nm lasers and a PlanApochromat 40× NA 1.3 objective (Zeiss) or a spinning disc confocal microscope CSU-W1 (Nikon/Andor) equipped with an iXON888 camera, using PlanApo 20× NA 0.75 and Apo LWD 40× NA 1.15 objectives. Maximal intensity projections of z-stacks were performed with ImageJ.

### ChIP

ChIP of histone modifications was performed from pInd11-shMLL1 Ls174T cell lines induced with 300 ng/ml doxycycline for 11 d, following the instructions of the iDEAL ChIP-seq kit for histones (Diagenode), as described in Grinat et al (2020). For ChIP of Mll1, pInd11-shMLL1 Ls174T and DLD1 cell lines were induced with 300 ng/ml doxycycline for 6 d and chromatin was prepared using the ChIP-IT Express kit (Active Motif). Cells were grown to 80% confluency, trypsinized for 3 min at 37°C, fixed in 1% formaldehyde for 10 min, and quenched in glycine for 5 min at RT. Chromatin was sheared with a Branson Sonifier 450 (3 min shearing time, duty cycle 60, output control 6, sonified 10× for histone ChIPs, 4× for Mll1 ChIPs, and 1 min pause between each sonication round). Shearing efficiency was checked on a 1% agarose gel. 10 μg of sheared chromatin were used for Mll1 ChIPs. ChIP-qPCR analysis was performed in a total volume of 20 μl SYBR green reaction mix (Roche Diagnostics) containing 0.25 μM of forward and reverse primers each in a CFX96-C1000T thermal cycler (Bio-Rad): 2 min at 50°C and 2 min at 95°C followed by 42 cycles of 15 s at 95°C and 1 min at 60°C. Ct values of precipitated DNA were calculated relative to input DNA (% input). ChIP-qPCR primers were designed using H3K4 methylation profiles available in the UCSC genome browser (human reference genome GRCh37/hg19) and the Mll1 ChIP-seq UCSC genome browser dataset from Active

Motif (https://www.activemotif.com/catalog/details/61295/mll-hrx-antibody-pab). Primer sequences used for ChIP-qPCR are given in Table S1.

## Western blotting

Organoids were harvested and washed once in ice-cold 0.1% BSA-PBS, and cells were washed twice in PBS before lysis in ice-cold RIPA buffer (50 mM Tris, pH 8.0, 150 mM NaCl, 0.1% SDS, 1% NP40, and 0.5% sodium deoxycholate) containing protease inhibitors (cOmplete Mini EDTA-free; Roche) and phosphatase inhibitor cocktails 2 and 3 (Sigma-Aldrich). Total cell extracts were separated on polyacrylamide gels and transferred to a nitrocellulose membrane via semidry transfer for 1 h 15 min at 90 mA. Membranes were blocked with 5% BSA or 5% skim milk in 0.1% Tween 20/TBS and probed with primary antibody diluted in blocking solution over night at 4°C. HRP-conjugated secondary antibodies were incubated for 1 h at RT. Immunoblots were developed with Western Lightning Plus ECL (PerkinElmer) for 3 min and imaged with a Vilber Lourmat imaging system FUSION SL-3.

## Antibodies

The following antibodies were used in this study (dilutions given for immunostaining): anti-Mll1 (D6G8N, #14197; Cell Signaling Technology, 1:100; 1:50 for ChIP), anti-E-cadherin (610181; BD, 1:200), anti-ChroA (Abcam, 1:300), anti-Mmp7 (Santa Cruz Biotechnology, 1:100), anti-GFP (ab6673; Abcam, 1:500), anti-Ki67 (MA5-14520; Thermo Fisher Scientific, 1:300), anti-BrdU (ab6326; Abcam, 1:100), anti-cleaved Caspase-3 (#9661; Cell Signaling Technology, RRID:AB_2341188, 1:400), anti-$\beta$-catenin (610153; BD, 1:300 for IHC, 1:1,000 for Western blot), anti-Lyz (A0099; DAKO, 1:500), anti-ITF (sc-18272; Santa Cruz Biotechnology, 1:300), anti-phospho-Erk1/2 (#4370; Cell Signaling Technology, RRID:AB_2315112, 1:200 for IHC), anti-phospho-Erk1/2 (M8159; Sigma-Aldrich, RRID:AB_477245, 1:3,000 for Western Blot), anti-Erk1/2 (9102; Cell Signaling Technology, RRID:AB_330744, 1:1,000 for Western Blot), anti-phospho-Mek1/2 (9154; Cell Signaling Technology, RRID:AB_2138017, 1:1,000 for Western Blot), anti-Mek1/2 (4694; Cell Signaling Technology, RRID:AB_10695868, 1:1,000 for Western Blot), anti-$\alpha$ tubulin (Sigma-Aldrich, 1:10,000 for Western Blot), anti-GATA6 (Cat. no. AF1700; R&D Systems, RRID:AB_2108901, 1:100–1:500), anti-GATA4 (sc-25310; Santa Cruz Biotechnology, RRID:AB_627667, 1:200), anti-H3K4me3 (#9727; Cell Signaling Technology, RRID:AB_561095, 1:50 for ChIP), anti-H3K27me3 (#07-449; Millipore, RRID:AB_310624, 5 $\mu$g for ChIP), and rabbit monoclonal IgG control (#3900; Cell Signaling Technology, RRID:AB_1550038). For immunofluorescence and IHC, cyanine-labelled secondary antibodies (Jackson ImmunoResearch) and HRP-conjugated polymer and DAB reagent (DAKO) were used.

## RNA preparation for RT–PCR analysis

Total RNA from organoids and snap-frozen tissue was isolated by Trizol extraction (Invitrogen) or with the NucleoSpin RNA isolation kit (Macherey-Nagel). DNA contaminations were removed by DNase1 digestion (Invitrogen) in the presence of RNase inhibitor (RNase Out; Invitrogen), and RNA was purified via phenol/chloroform extraction. For qRT-PCR, up to 5 $\mu$g of total RNA were reverse-transcribed with random hexamer primers (Invitrogen) and MMLV reverse transcriptase (200 U/$\mu$l; Promega). qRT-PCR was performed in a total volume of 20-$\mu$l SYBR green reaction mix (Roche Diagnostics) containing 0.25 $\mu$M of forward and reverse primers each in a CFX96-C1000T thermal cycler (Bio-Rad): 2 min at 50°C and 2 min at 95°C followed by 42 cycles of 15 s at 95°C and 1 min at 60°C. All reactions were performed as duplicates. Expression of target genes in treated versus control samples relative to the endogenous reference *GAPDH* was calculated using the $\Delta\Delta C_t$ method. Primer sequences used for qRT-PCR are listed in Table S2.

## Quantification and statistical analysis

All data are presented as mean ± SEM unless otherwise indicated. Statistical details of the experiments can be found in the figure legends. Graphs and statistics were generated with GraphPad Prism software. Tests for normal distribution were performed with D'Agostino-Pearson and Shapiro–Wilk tests. Significance (*P*-values) was determined with Mann–Whitney *U* test (two-tailed), two-tailed *t* test or ordinary one-way ANOVA. No statistical method was used to estimate sample size, and no specific randomization or blinding protocol was used. N indicates the numbers of independent biological replicates per experiment unless otherwise indicated. *P*-values ≤ 0.05 were considered statistically significant (\*$P$ ≤ 0.05, \*\*$P$ ≤ 0.01, \*\*\*$P$ ≤ 0.001, \*\*\*\*$P$ ≤ 0.0001).

# Data Availability

The dataset produced in this study is available in the following database: RNA seq data: Gene Expression Omnibus GSE177047.

# Supplementary Information

# Acknowledgements

We thank Walter Birchmeier for generous support of this study. We thank Marcel Harrig (MDC) for great reliability in maintaining the mouse colony, the Advanced Light Microscopy core facility of the MDC for assistance with imaging, H-P Rahn, and the FACS core facility of the MDC as well as the FACS core facility of the Biotechnology Center Dresden (Katja Schneider) for support with cell sorting. This work was supported by MDC central resources and by funding from Deutsche Forschungsgemeinschaft STE903/7-3 to AF Stewart and Si-1983/4-1 to M Sigal. J Grinat was funded in part by the Berlin School of Integrative Oncology of the Charité Medical School Berlin. The study was also supported by a generous donation in memoriam "Sören Piepgras" by Dr. MG.

## Author Contributions

J Grinat: formal analysis, validation, investigation, visualization, methodology, and writing—original draft, review, and editing.
F Kosel: investigation and methodology.
N Goveas: conducted the Paneth cell sort.

A Kranz: conducted the Paneth cell sort.

D Alexopoulou: performed bioinformatic analyses.

K Rajewsky: resources.

M Sigal: resources.

AF Stewart: resources.

J Heuberger: conceptualization, formal analysis, supervision, validation, investigation, visualization, methodology, project administration, and writing—original draft, review, and editing.

## Conflict of Interest Statement

The authors declare that they have no conflict of interest.

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
