## [Reviewer comments · Life Science Alliance]

Life Science Alliance

Epigenetic modifier balances Mapk and Wnt signaling in differentiation of goblet and Paneth cells

Johanna Grinat, Frauke Kosel, Neha Goveas, Andrea Kranz, Dimitra Alexopoulou, Klaus Rajewsky, Michael Sigal, A. Stewart, and Julian Heuberger

DOI: <https://doi.org/10.26508/lsa.202101187>

Corresponding author(s): Julian Heuberger, Charité - University Medicine Berlin

Review Timeline:

Submission Date:	2021-08-09
Editorial Decision:	2021-09-07
Revision Received:	2021-12-07
Editorial Decision:	2021-12-23
Revision Received:	2022-01-04
Accepted:	2022-01-05

Transaction Report:

September 7, 2021

Re: Life Science Alliance manuscript #LSA-2021-01187-T

Dr. Julian Heuberger

Charité

Augustenburger Platz 1, Medical Department, Division of Gastroenterology and Hepatology, Charité University Medicine

Berlin 13353

Germany

Dear Dr. Heuberger,

Thank you for submitting your manuscript entitled "Epigenetic modifier balances Mapk and Wnt signaling in differentiation of goblet and Paneth cells" to Life Science Alliance. The manuscript was assessed by expert reviewers, whose comments are appended to this letter. We invite you to submit a revised manuscript addressing the Reviewer comments.

Thank you for this interesting contribution to Life Science Alliance. We are looking forward to receiving your revised manuscript.

Sincerely,

Eric Sawey, PhD

Executive Editor

Life Science Alliance

<http://www.lsjournal.org>

B. MANUSCRIPT ORGANIZATION AND FORMATTING:

Reviewer #1 (Comments to the Authors (Required)):

In this paper, Grinat et al. provide evidence that the histone methyltransferase Mll1 plays a role in regulating Wnt- and Mapk-driven cell fate specification of intestinal secretory progenitors, namely Paneth and goblet cells. In particular, through conditional ablation of Mll1 in mice and intestinal organoids, the authors report that loss of Mll1 results in increase number of mixed lineage cells adopting both Paneth and goblet cell features. The authors further argue using mice constitutively expressing an active variant of Mek1, that this phenotype results from enhanced MAPK activity in Mll1 deficient cells that causes increased goblet cell differentiation but diminished proliferative potential of these cells. In conclusion the manuscript proposes a dual role for Mll1 in coordinating Wnt/Mapk-driven progenitor cell proliferation and differentiation.

Overall, the paper is well constructed, and conclusions are fairly supported by evidence provided. The manuscript is slightly derivative of the authors' previous work, which already had showed that loss of Mll1 in crypt overexpressing Beta-catenin causes enhanced Goblet/Paneth cell differentiation. The manuscript does provide novel insight into the role of Mll1 in regulating Mapk signalling activity in the context of elevated Wnt signalling but the precise mechanism by which Mll1 loss unleashes Mapk signalling is not addressed. Despite these weaknesses, commendable use of multiple loss and gain of function mouse models with appropriate reporter genes was presented. In conclusion the paper is suitable for Life Science Alliance once the following points have been addressed.

Specific comments

In their previous work the authors demonstrated that Mll1 is directly bound to ISC genes thereby activating their transcription. A similar mechanism might be proposed in this manuscript. The authors should examine whether key regulators of Paneth/Goblet cell differentiation (e.g. Spdef, Klf4 and Atoh etc.) are directly regulated by Mll1. At the very least this issue should be commented on in the discussion.

In Fig 1, the authors analysed cell type composition 50 days after tam induction. Although differential effects were observed, it is not clear why this particular timepoint was chosen. Were differences observed at earlier times post induction. Explanations should be given in the text.

The authors used the term "vesicle-containing" or "goblet-like" cells seen in Mll1^{-/-} mice crypts that were stained with Alcian blue (Fig. S1D). A better morphological description of these cells is required. Their analysis should include EM images showing the ultrastructure of these vesicle-containing cells.

In Fig 4, the authors examined pErk and Mek levels in Mll1 deficient cells overexpressing Beta-catenin. What are the effects of loss of Mll1 alone on Mapk signaling (i.e. Mll1 KO vs wt crypts)?

In Fig 4b, it is unclear whether the of levels p-Erk1/2 and total Erk1/2 were normalized to housekeeping (B-cat or tubulin). Should be specified in the figure legend. Also since a single set of housekeeping bands are presented, the same blot should be used to probe for both phospho- and total Erk.

Reviewer #2 (Comments to the Authors (Required)):

In this manuscript the authors report that the epigenetic regulator Mll1 regulate intestinal epithelium differentiation into goblet and paneth cells. Mechanistically, Mll1 coordinates Wnt and MAPK signals to maintain progenitor cell proliferation and specifies the fate of secretory paneth and goblet cells. Several knockout/Knockin mouse models were used, the results are convincing. The manuscript is generally well-written, and is helpful for researchers in the field of intestine homeostasis. The manuscript data can basically support every key point of the paper.

Major comments:

- 1) For Fig.1, the changes in duodenum, ileum and jejunum of Mll1^{-/-} mice and wild-type mice should be compared respectively
- 2) Fig. 2, organoid section staining for goblet and Paneth cells should be added.

- 3) Does the function of Mll1 depend on its methyltransferase activity? Methyltransferase inhibitors should be used to verify
- 4) Inhibitors of Wnt and MAPK signaling will increase the evidence of the manuscript
- 5) What is the effect of goblet/Paneth cell changes induced by Mll1^{-/-} on small intestinal function?
- 6) Is there any phenotype in colorectal epithelium in Mll1^{-/-} mice?

Reviewer #3 (Comments to the Authors (Required)):

Summary

In this article, Grinat and colleagues shed light into the specification of intestinal epithelial secretory lineage by epigenetic factors. In particular, they show that the methyltransferase Mll1 prevents intestinal stem cells from differentiating, while also being involved in goblet/Paneth cell specification. Mll1-deficient cells differentiate into the secretory lineage, but result in cells with a double Paneth/goblet phenotype, likely due to Mapk signalling activity. Although they do not delve into the molecular mechanism, it is a comprehensive phenotypical description of the loss-of-function / gain-of-function of the different pathways involved, both in vivo in histological sections of mouse intestine and organoids.

Main points of the paper

Ablation of Mll1 causes aberrant secretory differentiation in intestinal epithelial crypts.
Data are strongly supportive.

Mll1-deficient intestinal organoids show increased expression of secretory Paneth and goblet cell genes.

Data are strongly supportive of the increased expression of secretory Paneth and goblet cell genes. However, albeit non-significant for ChgA and Syp, there is also a decrease in Neurog3 expression. It could be mentioned in the discussion.

Mll1-deficient Wnt-high crypt cells exhibit goblet cell features.

Data are supportive, but it would benefit from showing some extra controls. For instance, the authors mention that there are less frequent ITF-positive goblet cells in the Mll1^{-/+} background upon beta-catenin GOF, however, b-cat wt background is not shown in parallel for comparison. Either this control is performed and quantified (Lyz, ITF, DAPI staining on β -cat-wt;Mll1^{+/-}; same as GOF but prior to tamoxifen administration) or if recombinant negative vs. positive crypts are to be compared in the same β -cat-GOF;Mll1^{+/-} sample, EGFP staining should be shown in consecutive sections (if not possible to stain on the same section).
Time frame 1-3 months.

I would also strongly recommend adding a control to Figs. S3E and S3H. However, I understand that adding control levels of expression of Axin2 in Fig. S3E would involve sorting and RNA-seq of b-cat wt samples and might be out of a reasonable time frame. Regarding controls in Fig. S3H, it would help to see the difference b-cat-GOF confers to organoids. Nuclear b-cat staining should also be improved in Fig. S3H, for example by using an anti-active-b-catenin specific antibody (such as Anti-Active- β -Catenin (Anti-ABC) Antibody 05-665 from Millipore). Time frame 1-3 months.

Loss of Mll1 unleashes Mapk signalling in Wnt-activated crypt cells.

Data are strongly supportive.

Mll1 restricts Mapk-driven goblet cell differentiation.

Data are strongly supportive.

Mll1 controls the Wnt/Mapk-driven specification of Paneth and goblet cells.

Data are supportive, but Lyz⁺ PCs emerging in the Mll1 null background also show some residual ITF staining, maybe it could be mentioned that the PC that appear retain some ITF expression. My concern is that it is observed also in control samples. Does this always happen in the wildtype background in organoids? Does this also happen in vivo in the wt background? Could this be an issue with the organoid immunofluorescence? I suggest doing a double immunofluorescence with both secondary antibodies but only one of the primary antibodies (and an unspecific antibody of the other's species) at a time, to confirm the absence of cross-reaction. Time frame 1-3 months.

Additional issues

I would like to see some discussion about the possible targets of Mll1 to mediate on one hand proliferation and on the other hand Paneth/goblet cell differentiation, maybe extracted from the transcriptional data from sorted PCs. I believe that performing a ChIP-seq in the transit amplifying cells or secretory progenitors would be out of the scope of the paper / reasonable time frame, but mention of the candidates that could be critical on the interplay of Wnt and Mapk pathways regulated by Mll1 would be interesting.

Minor comments: intersperse to interspersed in introduction (page 3).

LSA-2021-01187-T

Manuscript Grinat et al. "Epigenetic modifier balances Mapk and Wnt signalling in differentiation of goblet and Paneth cells" submitted to Life Science Alliance on LSA-2021-01187-T.

Point-by-point response to the reviewers

We thank all reviewers for their constructive suggestions on how to progress our work. We appreciate the overall positive opinion on our study. We have now addressed all questions and concerns. The additional data buttress the original findings from our study and strongly support our conclusions.

For clarity, all reviewer comments are shown in *italics*, with our responses below. **New parts of the manuscript that were included in the response are highlighted in yellow here as well as in the revised version of the manuscript.**

Reviewer #1:

In this paper, Grinat et al. provide evidence that the histone methyltransferase Mll1 plays a role in regulating Wnt- and Mapk-driven cell fate specification of intestinal secretory progenitors, namely Paneth and goblet cells. In particular, through conditional ablation of Mll1 in mice and intestinal organoids, the authors report that loss of Mll1 results in increase number of mixed lineage cells adopting both Paneth and goblet cell features. The authors further argue using mice constitutively expressing an active variant of Mek1, that this phenotype results from enhanced MAPK activity in Mll1-deficient cells that causes increased goblet cell differentiation but diminished proliferative potential of these cells. In conclusion, the manuscript proposes a dual role for Mll1 in coordinating Wnt/Mapk-driven progenitor cell proliferation and differentiation.

Overall, the paper is well constructed, and conclusions are fairly supported by evidence provided. The manuscript is slightly derivative of the authors' previous work, which already had showed that loss of Mll1 in crypt overexpressing Beta-catenin causes enhanced Goblet/Paneth cell differentiation. The manuscript does provide novel insight into the role of Mll1 in regulating Mapk signalling activity in the context of elevated Wnt signalling but the precise mechanism by which Mll1 loss unleashes Mapk signalling is not addressed. Despite these weaknesses, commendable use of multiple loss and gain of function mouse models with appropriate reporter genes was presented. In conclusion the paper is suitable for Life Science Alliance once the following points have been addressed.

Specific comments

In their previous work the authors demonstrated that Mll1 is directly bound to ISC genes thereby activating their transcription. A similar mechanism might be proposed in this manuscript. The authors should examine whether key regulators of Paneth/Goblet cell differentiation (e.g. Spdef, Klf4 and Atoh etc.) are directly regulated by Mll1. At the very least this issue should be commented on in the discussion.

We agree with the reviewer that mechanistic data will strengthen our manuscript. We have extended our study to include mechanistic data, which we now present in Figures 6 and S6 (pages 13-14, lines 321-362). Our data indicate that Mll1 regulates the goblet cell state rather than the expression of specific secretory cell markers. It controls secretory cell fate on a higher level, which is reflected in the altered expression of whole marker panels rather than specific marker genes in Mll1^{-/-} cells. In our previous studies we detected that Mll1 regulates Gata4 transcription factors (Grinat *et al*, 2020). Here we demonstrate that direct regulation of Gata4 expression by the Mll1 methyltransferase accounts for the role of Mll1 in controlling secretory cell fates and provides insight into the mechanism how Mll1 impacts on Mapk signalling.

Mll1^{-/-} crypts and organoids exhibit reduced levels of Gata4. Gata4 had previously been described to repress *Atoh1* expression (Kohlhofer *et al*, 2016; Beuling *et al*, 2011). In accordance, Mll1^{-/-} organoids showed increased levels of *Atoh1* expression. Using CRISPR/Cas9 in intestinal organoids, we demonstrate that loss of Gata4 increases Mapk signalling (p-Erk) and goblet cell differentiation, providing mechanistic explanation for the increased levels of Mapk activity and goblet-like cells in Mll1-deficient crypts and organoids. To further delineate the mechanistic details, we studied the human colon cell lines Ls174T and DLD1, from which we had established inducible MLL1 knockdown lines (Grinat *et al*, 2020). Using CRISPR/Cas9 knockout and overexpression of GATA6, the colon homologue of Gata4 (Whissell *et al*, 2014), we show that the Ls174T colon cells behave similarly to intestinal organoids with respect to the GATA6-ATO1 regulation. ChIP analyses in non-induced and doxycycline-induced shMLL1 knockdown cells revealed specific binding of MLL1 to the *GATA6* promoter. Activating H3K4me3 marks at the *GATA6* promoter switched to repressive H3K27me3 upon knockdown of MLL1, revealing the role of MLL1 enzymatic activity in *GATA6* regulation. Treatment of intestinal organoids with the methyltransferase inhibitor MI-2 decreased *Gata4* and increased goblet cell marker expression, corroborating the enzymatic role of MLL1 in controlling secretory cell fates through Gata4/6 transcription factors.

In Fig 1, the authors analysed cell type composition 50 days after tam induction. Although differential effects were observed, it is not clear why this particular timepoint was chosen. Were differences observed at earlier times post induction. Explanations should be given in the text.

The differential effects described in Fig. 1 were indeed clearly observed at 50 days after mutagenesis. Earlier time points were analysed and showed inceptive goblet/Paneth cell changes, but to a lesser extent than observed at day 50 (see Figure 1 below in this letter, in particular light Alcian Blue-positive staining in lower crypt cells and Paneth cells at the crypt bottom versus strong Alcian Blue positivity at day 50, Fig. 1A). Of note, the switch towards a mixed Paneth/goblet identity is very obvious early on in the background of high Wnt signalling (see analysis of day 10 β -cat^{GOF}; Mll1^{-/-} in Fig. 3, which was chosen to analyse the secretory lineage fate before the mice develop β -cat^{GOF}-driven tumors (Grinat *et al*, 2020). We have previously shown that Mll1 sustains the expression of its target genes (e.g. *Lgr5*), but is not responsible for their activation (Grinat *et al*, 2020). Hence, one may hypothesize an

epigenetic reprogramming of cell fate that is occurring upon loss of Mll1 and is generally slow but pushed in the presence of oncogenic or pro-differentiation signals such as high Wnt activation. This hypothesis corresponds with our organoid data in Fig. 2, where Wnt activation through Wnt3a treatment promotes Paneth/goblet cell differentiation of Mll1^{-/-} organoids.

We have now included an explanation for the chosen time point in the text (page 5, lines 114-115): “To exclude transient effects, we analysed mutant intestinal epithelia at 30 to 50 days after mutagenesis.”

Figure 1. Immunohistochemistry for Mll1 (left) and Alcian Blue staining for mucus-containing goblet cells (right) on serial sections of Mll1^{-/-} intestine at 10 days after induction of mutagenesis. Compare the presence of Alcian Blue-positive cells in the Mll1^{-/-} crypts (marked by asterisks) to adjacent non-recombined crypts.

The authors used the term "vesicle-containing" or "goblet-like" cells seen in Mll1^{-/-} mice crypts that were stained with Alcian blue (Fig. S1D). A better morphological description of these cells is required. Their analysis should include EM images showing the ultrastructure of these vesicle-containing cells.

We thank the reviewer for pointing this out. Unfortunately, due to the Covid-19 pandemic and staff changes in the host lab, the breeding of the Mll1^{fllox} mice had to be stopped and we are currently unable to obtain the new material required for the suggested EM analysis. However, we performed super-resolution microscopy of Lyz- and ITF-stained Mll1^{-/-} intestinal crypts (Figure 2 below in this letter). While the super-resolution images clearly show secretory vesicles in the Paneth cells, the double-positive cells do not exhibit clear vesicular structures. We agree with the reviewer that the use of the term “vesicle-containing” requires a more detailed morphological description of these cells. Initially, by choosing the terminology “vesicle-containing” or “goblet-like” we aimed to differentiate between the double-positive cells and pure Paneth and goblet cells. Indeed, it is an imprecise description and we therefore suggest to change the term “vesicle-containing” to “cells with goblet cell-like features” to refer to the observation in Fig. S1D (page 5, line 111) and “mucus-containing goblet-like cells” for Fig. 1A (page 5, lines 112-113). The labelling and legend of Fig. S1D were adjusted accordingly.

Figure 2: Super-resolution section through a *Mll1*-deficient intestinal crypt stained for the Paneth cell marker Lyz (green) and the goblet cell marker ITF (red). Nuclei stained with DAPI (blue). Grey arrows mark Paneth cells with secretory vesicles, white arrows indicate double-positive cells.

In Fig 4, the authors examined pErk and Mek levels in *Mll1* deficient cells overexpressing Beta-catenin. What are the effects of loss of *Mll1* alone on Mapk signaling (i.e. *Mll1* KO vs wt crypts)?

An immunohistochemistry staining for phospho-Erk1/2 in *Mll1*^{-/-} crypts is now included in the new Supplementary Fig. S4B (page 9, lines 219-221). In just *Mll1*-deficient crypts the increase in phospho-Erk1/2 levels is not detectable. However, the *Mll1*-deficient crypts exhibit increased numbers of goblet-like cells (Fig. 1A and new Fig. S4B), indicating that loss of *Mll1* alone enhances goblet cell differentiation, but its effect on Mapk signalling is not as strong as in crypts with hyperactivated Wnt signalling. By enhancing the Wnt effect, the effect of the loss of *Mll1* on the crypt differentiation signals becomes visible, which we also observed in organoid culture in Fig. 2 where Wnt3a treatment promotes Paneth/goblet cell differentiation of *Mll1*^{-/-} organoids. This might be attributed to an expansion of the secretory progenitor pool in Wnt-active crypts, as we have now included in the discussion (pages 15-16, lines 387-393): “Of note, the loss of *Mll1* results in increased goblet cell specification in the wild-type intestine but a global increase in Mapk activity is not visible, which we ascribe to the transient nature and small population of secretory progenitor cells, in which this regulatory circuit is active. Consistent with the role of Wnt signalling for Math1-positive secretory progenitors (Tian *et al*, 2015), Wnt activation by Wnt3a treatment or β -cat^{GOF} expands the secretory progenitor pool and hence allows to detect molecular and signalling effects.”

In Fig 4b, it is unclear whether the levels p-Erk1/2 and total Erk1/2 were normalized to housekeeping (*B-cat* or tubulin). Should be specified in the figure legend. Also since a single set of housekeeping bands are presented, the same blot should be used to probe for both phospho- and total Erk.

The levels of p-Erk1/2 in Fig. 4B were normalized to total Erk1/2, same for p-Mek1/2 to total Mek1/2. Both were not normalized to tubulin before. The housekeeping (tubulin) serves as a loading control for beta-catenin but not p-Erk1/2 and p-Mek1/2. We have clarified this now in the figure legend (page 37, line 913). The p-Erk1/2 and p-Mek1/2 parts of the blot originate from the same membrane that was cut in half after blotting. The

Figure 3: Quantification of total Erk1/2 and total Mek1/2 levels from two membrane parts (Fig. 4B) as proof of equal loading, (+/-) is β -cat^{GOF}; *Mll1*^{+/-} and (-/-) is β -cat^{GOF}; *Mll1*^{-/-}.

samples were prepared as a common master mix and run as duplicates on the same gel. The membrane parts were stained in a reciprocal way: p-Erk1/2 and total Mek1/2 on membrane part 1, p-Mek1/2 and total Erk1/2 on membrane part 2. We show on the right (Figure 3 in this letter) a quantification of total Mek1/2 and total Erk1/2 probed on the two different membrane parts, revealing equal levels between the compared samples (β -cat^{GOF}; Mll1^{+/-} and β -cat^{GOF}; Mll1^{-/-}) on the two membrane parts. Hence, the loading is equal and our quantification in Fig. 4B is valid. Moreover, our Western blotting data on increased p-Erk1/2 levels in Mll1-deficient β -cat^{GOF} organoids are backed up by the p-Erk1/2 immunohistochemistry in Fig. 4A.

Reviewer #2:

In this manuscript the authors report that the epigenetic regulator Mll1 regulate intestinal epithelium differentiation into goblet and paneth cells. Mechanistically, Mll1 coordinates Wnt and MAPK signals to maintain progenitor cell proliferation and specifies the fate of secretory paneth and goblet cells. Several knockout/Knockin mouse models were used, the results are convincing. The manuscript is generally well-written, and is helpful for researchers in the field of intestine homeostasis.

The manuscript data can basically support every key point of the paper.

Major comments:

1) For Fig.1, the changes in duodenum, ileum and jejunum of Mll1^{-/-} mice and wild-type mice should be compared respectively

We have now included a comparison of the changes in duodenum, ileum, jejunum and colon of Mll1^{-/-} and Mll1^{+/-} mice, the latter of which behave like wild-type mice (see Fig. S1D), by quantifying the number of secretory goblet-like cells (PAS-positive) in the crypts of the respective intestinal sections. Despite for the ileum, we detect a general increase in PAS-positive cells throughout the intestinal and colon epithelium in Mll1^{-/-} mice; the increase in goblet-like cells is most prominent in the jejunum, which we analyse in detail in our manuscript. The new data are included in Fig. S1E and pages 5-6, lines 115-118: “PAS staining of duodenum, jejunum, ileum and colon epithelial sections revealed increases in mucus-producing goblet cells in all parts of Mll1-deficient intestinal epithelia with the exception of the ileum (Fig. S1E). Increased goblet cell specification was most prominent in the jejunum.”

2) Fig. 2, organoid section staining for goblet and Paneth cells should be added.

We have now added goblet and Paneth cell stainings on wild-type and Mll1-deficient organoid sections (new Fig. S2A). As the RT-PCR data (Fig. 2B) suggest, Mll1-deficient organoids show increased numbers of goblet and slightly increased Paneth cells (page 7, lines 147-149 and page 42, lines 1018-1020).

3) Does the function of Mll1 depend on its methyltransferase activity? Methyltransferase inhibitors should be used to verify

We treated β -cat^{GOF} organoids with MI-2, which inhibits the interaction of Mll1 and the scaffold protein Menin and thereby impairs Mll1 methyltransferase activity (Grembecka *et al*, 2012; Heuberger *et al*, 2021). MI-2 treatment reduced the Wnt-induced expression of the Paneth cell marker *Mmp7* and reverted the Wnt-induced decrease in *Gob5* and by tendency *Muc2* expression (new Fig. 3D, page 9, lines 201-205). These data show that the function of Mll1 in this context depends on its methyltransferase activity, which is further corroborated by our new mechanistic data on the Mll1-dependent regulation of *Gata4/6* via H3K4me3 (new Fig. 6 and S6). Mll1^{-/-} crypts and organoids exhibit reduced levels of Gata4. Using CRISPR/Cas9 in human colon cells and intestinal organoids, we demonstrate that loss of Gata4/6 increases Mapk signalling (p-Erk) and goblet cell differentiation, providing a mechanistic explanation for the increased levels of Mapk activity and goblet-like cells in

Mll1-deficient crypts and organoids (Fig. 6, S6). Inhibition of Mll1 methyltransferase activity by MI-2 likewise reduced *Gata4* expression in organoids with a concomitant increase in goblet cell marker expression (Fig. 6J, page 14, lines 357-362), verifying that the function of Mll1 in Paneth and goblet cell specification depends on its methyltransferase activity.

4) *Inhibitors of Wnt and MAPK signaling will increase the evidence of the manuscript*

We thank the reviewer for this suggestion to broaden the experimental evidence of our manuscript with inhibitor studies. While we strongly agree that the methyltransferase inhibitor studies were critical to understand the role of Mll1 in Paneth-goblet cell specification (see Question 3 above), the effects of Wnt and Mapk inhibition in intestinal organoids have been published. While Mapk inhibition shifts intestinal cell fate clearly towards the Paneth cell lineage on the expense of goblet cells (Heuberger *et al*, 2014), Wnt inhibition shifts cell fate specification towards goblet cells and the combination of Wnt and Mapk inhibition promotes the enteroendocrine cell fate, in combination with inhibition of Notch signalling (Basak *et al*, 2017). The genetic evidence presented in our manuscript is in agreement with the published inhibitor data.

5) *What is the effect of goblet/Paneth cell changes induced by Mll1^{-/-} on small intestinal function?*

Loss of Mll1 in our study had no effect on mouse survival or fitness. We did not observe any effect on small intestinal function at homeostasis. Our lineage tracing experiments demonstrated fully recombined crypt-villus units without impaired morphology (Fig. S1A). However, it is important to note that for the *in vivo* studies we used the *Lgr5*-Cre^{ERT2} mouse strain, which is known to be patchy and hence leads to mosaic ablation of Mll1 in the intestinal epithelium. This mosaic loss of Mll1 is sufficient to study cell fate specification, but might mask functional consequences for the gut. In Goveas *et al*. (2021), we observed reduced survival probability upon ablation of Mll1 in all crypts of the intestinal epithelium using the *Villin*-Cre^{ERT2}. Whether this is attributed to goblet/Paneth cell changes is unclear. The ablation of Mll1 in all cell types of intestinal tissue using the *Rosa26*-Cre^{ERT2} resulted in intestinal failure (Goveas *et al*, 2021), indicating that Mll1 plays an additional role in other than epithelial cells, which is required for intestinal function.

6) *Is there any phenotype in colorectal epithelium in Mll1^{-/-} mice?*

We have now analysed secretory goblet-like cell numbers in the colon epithelium of Mll1^{-/-} mice and found a slight increase in PAS-positive cells in epithelia lacking Mll1 (now included in Fig. S1E). As our manuscript focusses on Paneth-goblet cell differentiation, we chose the small intestine (jejunum) as our model section. The colon epithelium does not possess Paneth cells and cell fate regulation might differ from the Mll1/Wnt/Mapk-driven regulation that we observe in the small intestine.

Reviewer #3:

Summary

In this article, Grinat and colleagues shed light into the specification of intestinal epithelial secretory lineage by epigenetic factors. In particular, they show that the methyltransferase Mll1 prevents intestinal stem cells from differentiating, while also being involved in goblet/Paneth cell specification. Mll1-deficient cells differentiate into the secretory lineage, but result in cells with a double Paneth/goblet phenotype, likely due to Mapk signalling activity. Although they do not delve into the molecular mechanism, it is a comprehensive phenotypical description of the loss-of-function / gain-of-function of the different pathways involved, both in vivo in histological sections of mouse intestine and organoids.

Main points of the paper

Ablation of Mll1 causes aberrant secretory differentiation in intestinal epithelial crypts.

Data are strongly supportive.

Mll1-deficient intestinal organoids show increased expression of secretory Paneth and goblet cell genes.

Data are strongly supportive of the increased expression of secretory Paneth and goblet cell genes. However, albeit non-significant for ChgA and Syp, there is also a decrease in Neurog3 expression. It could be mentioned in the discussion.

We have now included discussion of the slightly decreased *Neurog3* expression in Mll1-deficient organoids: “Mll1^{-/-} organoids showed a slightly reduced expression of the enteroendocrine progenitor marker *Neurog3*, which may further promote the acquisition of a goblet cell fate (Li et al, 2021), but the number of differentiated enteroendocrine cells was unchanged in Mll1^{-/-} intestine” (pages 16-17, lines 417-420).

Mll1-deficient Wnt-high crypt cells exhibit goblet cell features.

Data are supportive, but it would benefit from showing some extra controls. For instance, the authors mention that there are less frequent ITF-positive goblet cells in the Mll1^{-/+} background upon beta-catenin GOF, however, b-cat wt background is not shown in parallel for comparison. Either this control is performed and quantified (Lyz, ITF, DAPI staining on β -cat-wt;Mll1^{+/-}; same as GOF but prior to tamoxifen administration) or if recombinant negative vs. positive crypts are to be compared in the same β -cat-GOF;Mll1^{+/-} sample, EGFP staining should be shown in consecutive sections (if not possible to stain on the same section). Time frame 1-3 months.

To corroborate our statement that β -cat^{GOF}; Mll1^{+/-} intestine exhibits less frequent goblet cells, we have now included Alcian Blue stainings of non-recombined (wild-type) and β -cat^{GOF} intestine (new Fig. S3C). These data confirm that there are less goblet cells in the Wnt-high β -cat^{GOF} background. As the crypt-villus architecture is lost in β -cat^{GOF} intestine (tumors) at the time point of our analysis, a quantification comparing the numbers of goblet cells on a per-crypt basis is not possible. Given that high Wnt activity is known to reduced

goblet cell differentiation (Sansom *et al*, 2004), we suggest to keep it a qualitative comparison. Unlike recombined crypts in the $Mek1^{GOF}$ mice (Fig. 4), mutant crypts in $\beta\text{-cat}^{GOF}$; $Mll1^{+/-}$ mice are not marked by EGFP. Hence, the second suggestion to compare negative vs positive crypts in the same $\beta\text{-cat}^{GOF}$; $Mll1^{+/-}$ sample and identify recombined crypts via EGFP staining is experimentally not possible. As the Alcian Blue stainings that we included do not analyse specifically ITF-positive goblet cells, we changed the text accordingly: “Indeed, the high Wnt activity imposed a Paneth-like identity on the epithelial cells: $\beta\text{-cat}^{GOF}$; $Mll1^{+/-}$ intestines showed high numbers of Mmp7- and Lyz-positive cells, and few ITF-positive goblet cells (Fig. S3B). Alcian Blue staining of control and $\beta\text{-cat}^{GOF}$ intestines confirmed reduced numbers of goblet cells in Wnt-high epithelium (Fig. S3C).” (pages 7-8, lines 171-175).

I would also strongly recommend adding a control to Figs. S3E and S3H. However, I understand that adding control levels of expression of Axin2 in Fig. S3E would involve sorting and RNA-seq of b-cat wt samples and might be out of a reasonable time frame. Regarding controls in Fig. S3H, it would help to see the difference b-cat-GOF confers to organoids. Nuclear b-cat staining should also be improved in Fig. S3H, for example by using an anti-active-b-catenin specific antibody (such as Anti-Active- β -Catenin (Anti-ABC) Antibody 05-665 from Millipore). Time frame 1-3 months.

In Fig. S3F (previous S3E) we are investigating the effect that loss of *Mll1* has on Paneth cells. The *Axin2* expression is included to show that loss of *Mll1* does not globally reduce the expression of Wnt target genes, as we have previously established (Grinat *et al*, 2020). Regarding Fig. S3I (previous S3H), the use of the suggested anti-active- β -catenin antibody is not helping as the exon 3, which harbours the phosphorylation sites Ser37 or Thr41 recognized by this antibody, is deleted in the $\beta\text{-cat}^{GOF}$ mouse (deltaExon3, aa 5-80 (Harada *et al*, 1999)). However, we have now improved the staining for nuclear β -catenin with our anti- β -catenin antibody (BD 610153) and included a non-recombined control (Fig. S3I). Compared to control organoids, $\beta\text{-cat}^{GOF}$ organoids acquire a cauliflower-like shape and show high levels of nuclear β -catenin, which is unchanged upon ablation of *Mll1* (Fig. S3I).

*Loss of Mll1 unleashes Mapk signalling in Wnt-activated crypt cells.
Data are strongly supportive.*

*Mll1 restricts Mapk-driven goblet cell differentiation.
Data are strongly supportive.*

*Mll1 controls the Wnt/Mapk-driven specification of Paneth and goblet cells.
Data are supportive, but Lyz+ PCs emerging in the Mll1-null background also show some residual ITF staining, maybe it could be mentioned that the PC that appear retain some ITF expression. My concern is that it is observed also in control samples. Does this always happen in the wildtype background in organoids? Does this also happen in vivo in the wt background? Could this be an issue with the organoid immunofluorescence? I suggest doing*

a double immunofluorescence with both secondary antibodies but only one of the primary antibodies (and an unspecific antibody of the other's species) at a time, to confirm the absence of cross-reaction. Time frame 1-3 months.

Since Paneth and goblet cells derive from a common progenitor, a rarely detectable transient state *in vivo* cannot be excluded. We are aware of the double-positive nature of Paneth/goblet cells in Mll1-null background and mentioned it in the discussion: “While β -cat^{GOF}-induced Wnt activation promoted a Paneth cell fate and prevented Mapk-induced goblet cell differentiation, the ablation of Mll1 in β -cat^{GOF} and in β -cat^{GOF}; Mek1^{GOF} cells led to re-appearance of goblet cells, **largely as mixed Paneth-goblet entities**” (page 15, lines 385-386). We now also included a mention in the results section that Paneth cells appearing in Wnt/Mapk-high Mll1^{-/-} organoids retain some ITF expression: “Remarkably, Paneth and goblet cells re-appeared in organoids with homozygous ablation of Mll1, **largely as double-positive Paneth-goblet entities**, as shown by immunohistochemistry for Lyz, Mmp7 and ITF (Fig. 5A lower panel, quantification in B)” (page 12, lines 284-285).

To exclude that the double-positive cells we observe are due to a cross-reaction issue in the organoid immunofluorescence, we have now performed the suggested control staining (see Figure 4 below in this letter) and can exclude antibody cross-reaction.

Figure 4: Double immunofluorescence stainings on control, β -cat^{GOF}; Mek1^{GOF}; Mll1^{+/-} and β -cat^{GOF}; Mek1^{GOF}; Mll1^{-/-} organoid sections with specific primary antibody Lyz (rabbit, green) and unspecific

primary antibody SMA (goat, red). Single SMA-positive cell in bottom row demonstrates successful staining of the SMA antibody. There is no red staining in the Lyz+ cells, demonstrating the absence of cross-reaction of the secondary antibodies. DAPI stains nuclei (blue).

Additional issues

I would like to see some discussion about the possible targets of Mll1 to mediate on one hand proliferation and on the other hand Paneth/goblet cell differentiation, maybe extracted from the transcriptional data from sorted PCs. I believe that performing a ChIP-seq in the transit amplifying cells or secretory progenitors would be out of the scope of the paper / reasonable time frame, but mention of the candidates that could be critical on the interplay of Wnt and Mapk pathways regulated by Mll1 would be interesting.

We have included new mechanistic data that give insights into how Mll1 regulates Paneth/goblet cell differentiation (new Fig. 6, S6). Our data indicate that loss of Mll1 has an effect on cell state, associated with changes in Wnt/Mapk signalling, rather than a direct signalling effect. We unravel a molecular circuit around Gata4, which is directly regulated by the Mll1 methyltransferase. Mll1 sustains Gata4 expression, which represses Atoh1 and secretory differentiation, thus maintaining a progenitor cell state and cell proliferation. Wnt signalling is crucial for the secretory progenitor state and the Wnt/Mapk activity balances the cell fate decision – when Wnt signalling is high, secretory progenitor cells fate towards the Paneth cell state (Pinto *et al*, 2003; Heuberger *et al*, 2014). Secretory progenitor cells that downregulate Gata4 increase Atoh1 expression and Mapk activity, which pushes them towards differentiation and a goblet cell fate. Loss of Gata4 expression by Mll1 deficiency in crypts/organooids misbalances the cell fate acquisition at the secretory progenitor state and perturbs secretory cell fate specification. The discussion now includes these mechanistic findings (pages 16-17, lines 412-434).

Minor comments: intersperse to interspersed in introduction (page 3).

We changed intersperse to interspersed (page 3, line 51).

References

- Basak O, Beumer J, Wiebrands K, Seno H, van Oudenaarden A & Clevers H (2017) Induced Quiescence of Lgr5+ Stem Cells in Intestinal Organoids Enables Differentiation of Hormone-Producing Enteroendocrine Cells. *Cell Stem Cell* 20: 177-190 e4
- Beuling E, Baffour-Awuah NY, Stapleton KA, Aronson BE, Noah TK, Shroyer NF, Duncan SA, Fleet JC & Krasinski SD (2011) GATA factors regulate proliferation, differentiation, and gene expression in small intestine of mature mice. *Gastroenterology* 140: 1219–1229
- Goveas N, Waskow C, Arndt K, Heuberger J, Zhang Q, Alexopoulou D, Dahl A, Birchmeier W, Anastassiadis K, Stewart AF, *et al* (2021) MLL1 is required for maintenance of intestinal stem cells. *PLOS Genet* 17: e1009250
- Grembecka J, He S, Shi A, Purohit T, Muntean AG, Sorenson RJ, Showalter HD, Murai MJ, Belcher AM, Hartley T, *et al* (2012) Menin-MLL inhibitors reverse oncogenic activity of MLL fusion proteins in leukemia. *Nat Chem Biol* 8: 277–284
- Grinat J, Heuberger J, Vidal RO, Goveas N, Kosel F, Berenguer-Llargo A, Kranz A, Wulf-Goldenberg A, Behrens D, Melcher B, *et al* (2020) The epigenetic regulator Mll1 is required for Wnt-driven intestinal tumorigenesis and cancer stemness. *Nat Commun* 11: 6422
- Harada N, Tamai Y, Ishikawa T, Sauer B, Takaku K, Oshima M & Taketo MM (1999) Intestinal polyposis in mice with a dominant stable mutation of the beta-catenin gene. *EMBO J* 18: 5931–5942
- Heuberger J, Grinat J, Kosel F, Liu L, Kunz S, Vidal RO, Keil M, Haybaeck J, Robine S, Louvard D, *et al* (2021) High Yap and Mll1 promote a persistent regenerative cell state induced by Notch signaling and loss of p53. *Proc Natl Acad Sci* 118: e2019699118
- Heuberger J, Kosel F, Qi J, Grossmann KS, Rajewsky K & Birchmeier W (2014) Shp2/MAPK signaling controls goblet/paneth cell fate decisions in the intestine. *Proc Natl Acad Sci U S A* 111: 3472–3477
- Kohlhofer BM, Thompson CA, Walker EM & Battle MA (2016) GATA4 regulates epithelial cell proliferation to control intestinal growth and development in mice. *Cell Mol Gastroenterol Hepatol* 2: 189–209
- Pinto D, Gregorieff A, Begthel H & Clevers H (2003) Canonical Wnt signals are essential for homeostasis of the intestinal epithelium. *Genes Dev* 17: 1709–1713
- Sansom OJ, Reed KR, Hayes AJ, Ireland H, Brinkmann H, Newton IP, Battle E, Simon-Assmann P, Clevers H, Nathke IS, *et al* (2004) Loss of Apc in vivo immediately perturbs Wnt signaling, differentiation, and migration. *Genes Dev* 18: 1385–1390
- Tian H, Biehs B, Chiu C, Siebel CW, Wu Y, Costa M, de Sauvage FJ & Klein OD (2015) Opposing activities of Notch and Wnt signaling regulate intestinal stem cells and gut homeostasis. *Cell Rep* 11: 33–42
- Whissell G, Montagni E, Martinelli P, Hernando-Momblona X, Sevillano M, Jung P, Cortina C, Calon A, Abuli A, Castells A, *et al* (2014) The transcription factor GATA6 enables self-renewal of colon adenoma stem cells by repressing BMP gene expression. *Nat Cell Biol* 16: 695–707

December 23, 2021

RE: Life Science Alliance Manuscript #LSA-2021-01187-TR

Dr. Julian Heuberger
Charité - University Medicine Berlin
Augustenburger Platz 1, Medical Department, Division of Gastroenterology and Hepatology, Charité University Medicine
Berlin 13353
Germany

Dear Dr. Heuberger,

Thank you for submitting your revised manuscript entitled "Epigenetic modifier balances Mapk and Wnt signaling in differentiation of goblet and Paneth cells". We would be happy to publish your paper in Life Science Alliance pending final revisions necessary to meet our formatting guidelines.

- please upload your tables as single files in editable .docx or .xls format
- please correct the typo mistake in your title in the manuscript text
- please add a callout for Figure S6H to your main manuscript text
- Supplementary References should be incorporated into the main References

A. FINAL FILES:

B. MANUSCRIPT ORGANIZATION AND FORMATTING:

Sincerely,

Reviewer #1 (Comments to the Authors (Required)):

The authors have satisfactorily addressed all my comments. I recommend publication of this manuscript.

Reviewer #2 (Comments to the Authors (Required)):

The additional data is welcome and the authors addressed the concerns.

Reviewer #3 (Comments to the Authors (Required)):

Authors have addressed all raised issues. The quality of the manuscript has been significantly improved by adding insight into the molecular mechanism by which Mll1 regulates Gata4/6 and goblet cell state specification. I thank the authors for performing these informative experiments, some of the requested controls and improvement of the pictures. I recommend to accept the paper in its current form.

Minor comment: Line 539 change "supernatents" to "supernatants"

January 5, 2022

RE: Life Science Alliance Manuscript #LSA-2021-01187-TRR

Dr. Julian Heuberger
Charité - University Medicine Berlin
Augustenburger Platz 1, Medical Department, Division of Gastroenterology and Hepatology, Charité University Medicine
Berlin 13353
Germany

Dear Dr. Heuberger,

Thank you for submitting your Research Article entitled "Epigenetic modifier balances Mapk and Wnt signaling in differentiation of goblet and Paneth cells". It is a pleasure to let you know that your manuscript is now accepted for publication in Life Science Alliance. Congratulations on this interesting work.

DISTRIBUTION OF MATERIALS:

Again, congratulations on a very nice paper. I hope you found the review process to be constructive and are pleased with how the manuscript was handled editorially. We look forward to future exciting submissions from your lab.

Sincerely,
